# Direct reprogramming of human fibroblasts into insulin-producing cells using transcription factors

Marta Fontcuberta-PiSunyer[1], Ainhoa García-Alamán[1,2], Èlia Prades [1], Noèlia Téllez [2,3,4], Hugo Alves-Figueiredo[1,13], Mireia Ramos-Rodríguez [5], Carlos Enrich[1,6], Rebeca Fernandez-Ruiz[1,2], Sara Cervantes[1], Laura Clua[7], Javier Ramón-Azcón [7,8], Christophe Broca[9], Anne Wojtusciszyn[9,10], Nuria Montserrat [7,8,11], Lorenzo Pasquali[5], Anna Novials[1,2], Joan-Marc Servitja[1,2], Josep Vidal[1,2,6,12], Ramon Gomis[1,2,6] & Rosa Gasa [1,2✉]

Direct lineage reprogramming of one somatic cell into another without transitioning through a progenitor stage has emerged as a strategy to generate clinically relevant cell types. One cell type of interest is the pancreatic insulin-producing β cell whose loss and/or dysfunction leads to diabetes. To date it has been possible to create β-like cells from related endodermal cell types by forcing the expression of developmental transcription factors, but not from more distant cell lineages like fibroblasts. In light of the therapeutic benefits of choosing an accessible cell type as the cell of origin, in this study we set out to analyze the feasibility of transforming human skin fibroblasts into β-like cells. We describe how the timed-introduction of five developmental transcription factors (Neurog3, Pdx1, MafA, Pax4, and Nkx2-2) promotes conversion of fibroblasts toward a β-cell fate. Reprogrammed cells exhibit β-cell features including β-cell gene expression and glucose-responsive intracellular calcium mobilization. Moreover, reprogrammed cells display glucose-induced insulin secretion in vitro and in vivo. This work provides proof-of-concept of the capacity to make insulin-producing cells from human fibroblasts via transcription factor-mediated direct reprogramming.

[1] Institut d'Investigacions Biomèdiques August Pi i Sunyer (IDIBAPS), Barcelona, Spain. [2] CIBER de Diabetes y Enfermedades Metabólicas Asociadas, Instituto de Salud Carlos III, Madrid, Spain. [3] Faculty of Medicine of University of Vic, Central University of Catalonia (UVic-UCC), Vic, Spain. [4] Institute of Health Research and Innovation at Central Catalonia (IRIS-CC), Vic, Spain. [5] Department of Medicine and Life Sciences, Universitat Pompeu Fabra, Barcelona, Spain. [6] Faculty of Medicine and Health Sciences, Universitat de Barcelona, Barcelona, Spain. [7] Institute for Bioengineering of Catalonia (IBEC), The Barcelona Institute of Technology (BIST), Barcelona, Spain. [8] Catalan Institution for Research and Advanced Studies (ICREA), Barcelona, Spain. [9] CHU Montpellier, Laboratory of Cell Therapy for Diabetes (LTCD), Hospital St-Eloi, Montpellier, France. [10] Service of Endocrinology, Diabetes and Metabolism, Lausanne University Hospital, Lausanne, Switzerland. [11] CIBER de Bioingeniería, Biomateriales y Nanomedicina, Instituto de Salud Carlos III, Madrid, Spain. [12] Endocrinology and Nutrition Department, Hospital Clinic of Barcelona, Barcelona, Spain. [13] Present address: Tecnológico de Monterrey, Escuela de Medicina y Ciencias de la Salud, Monterrey, N.L., México. ✉email: rgasa@recerca.clinic.cat

Direct lineage reprogramming entails the direct conversion of one differentiated cell type into another bypassing an intermediate pluripotent stage. This strategy is often based on the forced expression of cocktails of transcription factors that function as potent fate determinants during development of the cell type of interest[1–3]. As the number of cell types produced through direct conversion has rapidly increased in recent years, this strategy has emerged as a possible method for creating cell types with potential for use in therapeutic settings.

Pancreatic beta (β) cells produce insulin, which controls whole body glucose homeostasis. Diabetes is characterized by a relative or total lack of functional β cells, and cell replacement therapy has consequently emerged as a promising therapeutic option to treat and ultimately cure this disease. One of the strategies pursued to produce replacement β cells has been direct lineage reprogramming. A major breakthrough in this area was the discovery that three developmental transcription factors, namely Pdx1, Neurog3, and MafA, promoted the in situ conversion of acinar cells into insulin-producing cells in the mouse pancreas[4,5]. Since then, studies have shown that various combinations of these and other transcription factors can promote conversion toward a β-like fate in other pancreatic cell lineages, including ductal and glucagon-expressing α-cells, and in extra-pancreatic related endodermal cell lineages, such as liver, gallbladder, and gastrointestinal tract cells[6–11].

One of the most important aspects of direct reprogramming strategies is the choice of the cell source, especially when taking into account their clinical application. The initial material should ideally be available, simple to handle and to grow in the laboratory. In this regard, skin fibroblasts have been the preferred cell source for many reprogramming protocols and they have so far been successfully transformed into a variety of somatic cell types including cardiomyocytes[12], chondrocytes[13], neurons[14], oligodendrocyte progenitors[15], hepatocytes[16] or endothelial cells[17]. However, research to date suggests that fibroblasts are resistant to being transformed into β-like cell using lineage-specific transcription factors[4,9,18].

In light of the expanding number of cell types produced by direct lineage conversion procedures and the therapeutic interest of insulin-producing cells, we chose to thoroughly consider the viability of using fibroblasts as cells of origin in direct reprogramming protocols to generate β-like cells. Here we present a protocol based on a cocktail of five endocrine transcription factors that induces human fibroblasts to activate the β-cell transcriptional program while downregulating their native fibroblastic transcriptional program, resulting in the generation of cells that produce and secrete insulin in vitro and in vivo. We believe these findings demonstrate the feasibility of this approach and set the basis to further explore this alternative path for generation of β-like cells for disease modeling and cellular therapy.

## Results

### Exogenous expression of the transcription factors Pdx1, Neurog3, and MafA in human fibroblasts.
We first sought to examine whether the transcription factors Neurog3, Pdx1, and MafA could induce expression of the INSULIN (INS) gene in human fibroblasts as readout of the capacity of these cells to be transformed toward a β-cell fate. To deliver these factors we employed a polycistronic adenoviral vector carrying the three transgenes (Ad-NPM hereafter), which had been previously used to promote β-cell reprogramming from pancreatic acinar cells[19]. After optimization of adenoviral transduction in fibroblasts (see Methods), abundant (>80%) cells positive for Cherry, which is also encoded by Ad-NPM, were easily observable three and seven days after viral infection (Fig. 1a). Likewise, high levels of

transcripts encoding the NPM factors were expressed at both time points (Fig. 1b). Three days after addition of Ad-NPM we detected marginal levels of INS mRNA that were increased >10-fold by day 7 (Fig. 1c). As cell culture formulations can have a major impact on gene expression events and cellular reprogramming, we tested different conditions after Ad-NPM infection. We observed that moving to RPMI-1640 and, to a lesser extent, CMRL-1066 medium and lowering the fetal calf serum concentration to 6% dramatically boosted INS gene activation, reaching values that were 0.12% those of human islets (Fig. 1d and Supplementary Figure 1). Under the same culture conditions, only a very marginal induction of the INS gene occurred when the N + P + M factors were delivered simultaneously via distinct adenoviruses to human fibroblasts (Supplementary Fig. 2). In addition to INS, we discovered that the NPM factors also activated the hormone genes GLUCAGON (GCG) and SOMATOSTATIN (SST), albeit at lower levels than INS as indicated by decreased relative expression values (compared to the housekeeping gene TBP) (Fig. 1e). The NPM factors also induced expression of genes encoding islet differentiation transcription factors including NEUROD1, INSM1, PAX4, NKX2-2, and ARX (Fig. 1f).

To further establish if the NPM factors promoted cell fate conversion and not simply activated their target genes in fibroblasts, we surveyed expression of genes associated with the fibroblastic signature, including several factors involved in maintenance of the fibroblastic transcriptional network such as TWIST2, PRRX1, and LHX9[20]. We found that these genes were downregulated as early as three days after NPM introduction. Other fibroblast markers exhibited a more delayed response but, by day 7 post-NPM, all tested genes exhibited significant downregulation (Fig. 1g). Together, these experiments validate that islet cell fate can be induced in human fibroblasts using a defined set of transcription factors.

### Addition of exogenous Pax4 and Nkx2-2 after the NPM reprogramming cocktail in human fibroblasts.
The observed induction of the islet hormone genes GCG and SST implied that the NPM factors might not specifically endorse β-cell fate in fibroblasts. Furthermore, we found that these factors did not induce NKX6-1, which encodes a β-cell specific factor required for the formation of pancreatic β cells during development[21] and key for optimal maturation of stem cell β cells in vivo[22,23] (Fig. 2a). These findings indicated that the NPM factors suboptimally promoted a β-cell state in human fibroblasts. In order to enhance β-cell fate over other islet cell identities, we opted to add new transcription factors to the reprogramming cocktail.

Pax4 is activated downstream of Neurog3 during development[24] and has been shown to favor β- over α-cell specification[25,26], and to contribute to maintenance of the expression of Nkx6.1 in differentiating β cells[27]. Despite that the NPM factors induced endogenous PAX4 mRNA, the expression levels attained might not be sufficient to endorse β- over α-cell fate. Hence, we treated fibroblasts with an adenovirus encoding Pax4 three days after NPM (Fig. 2a). This resulted in the significant enhancement of INS expression as compared to NPM alone but, unexpectedly, GCG expression was also increased (Fig. 2a), indicating that ectopic Pax4 improved islet hormone gene expression without apparent impact on β- versus α-cell fate conversion in human fibroblasts.

As the NKX6-1 gene remained silent in response to NPM + Pax4 (Fig. 2a), we tried directly adding Nkx6-1 to the NPM reprogramming cocktail. However, exogenous Nkx6-1 resulted in considerable cell death irrespective of level of expression or timing of introduction. As an alternate approach, we added

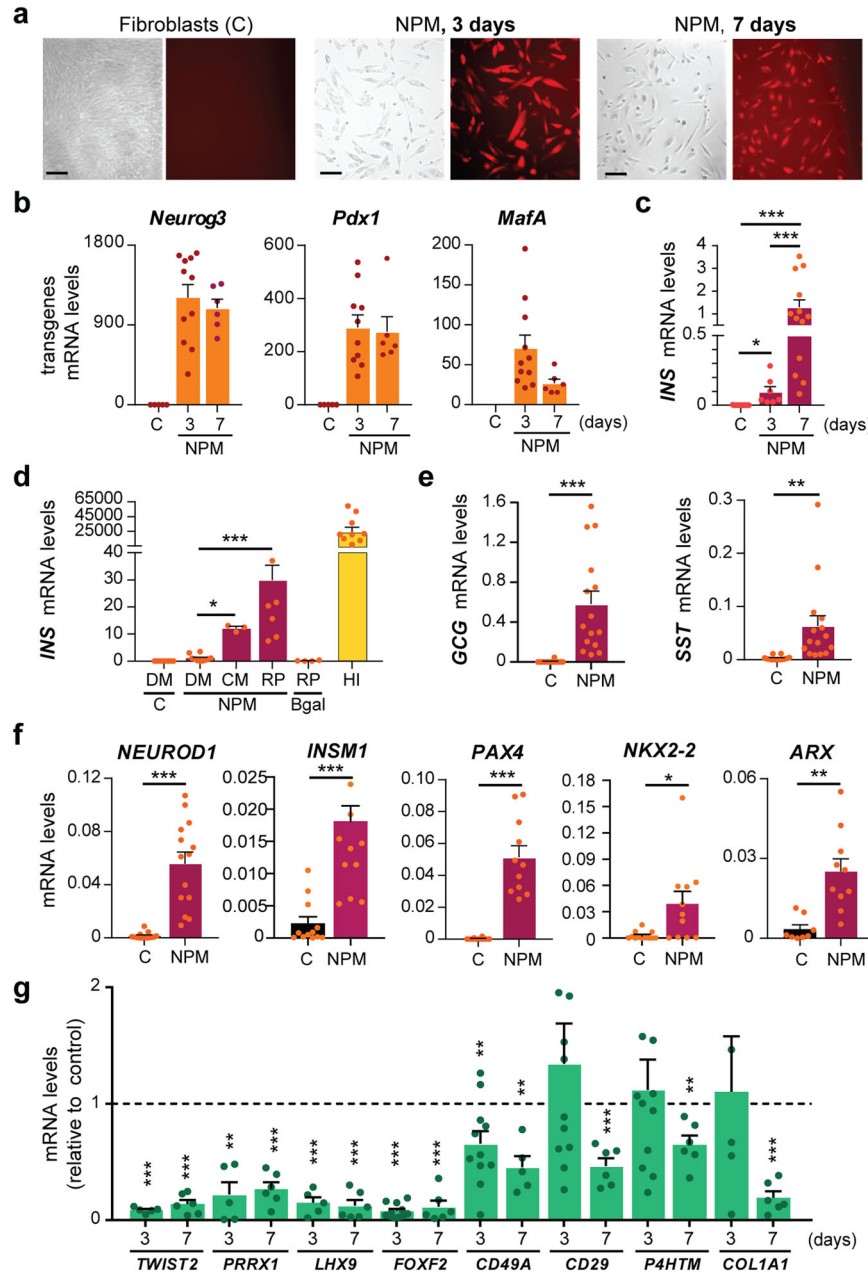

**Fig. 1 Introduction of the transcription factors Neurog3, Pdx1, and MafA activates pancreatic endocrine gene expression in human fibroblasts.** Human fibroblasts (HFF1) were infected with a polycistronic recombinant adenovirus encoding the transcription factors Neurog3, Pdx1, MafA, and the reporter protein Cherry (Ad-NPM). Untreated parental fibroblasts were used as controls (indicated as C in graphs). **a** Bright field images and Cherry immunofluorescence of control fibroblasts and fibroblasts infected with Ad-NPM at day 3 and 7 post-infection. Scale bar, 100 μm. **b** qPCR of transgenes at day 3 ($n = 11$) and 7 ($n = 6$) after infection with Ad-NPM. **c** qPCR of human *INS* at day 3 ($n = 7$) and 7 ($n = 13$) after infection with Ad-NPM. **d** qPCR of human *INS* in fibroblasts maintained in the indicated culture media (DM = DMEM; CM = CMRL; RP = RPMI) during 7 days after infection with Ad-NPM or with an adenovirus expressing B-galactosidase (B-gal) ($n = 3–10$). In yellow, *INS* mRNA levels in isolated human islets ($n = 10$). **e**, **f** qPCR of islet hormone genes (*GCG*, *SST*) and islet differentiation transcription factors (*NEUROD1, INSM1, PAX4, NKX2-2, ARX*) at day 7 post-NPM ($n = 8–15$). **g** qPCR of the indicated fibroblast markers at day 3 ($n = 5–11$) and day 7 post-NPM ($n = 5–6$). In **b–f**, expression levels are expressed relative to *TBP*. In **g**, expression is expressed relative to control fibroblasts, given the value of 1 (dotted line). Data are presented as the mean ± SEM for the number of samples indicated in parentheses. *$P < 0.05$; **$P < 0.01$; ***$P < 0.001$, between indicated conditions using unpaired *t*-test (**b–f**), or relative to control fibroblasts using one sample *t*-test (**g**).

exogenous Nkx2-2, which also regulates early β-cell differentiation and is an upstream activator of Nkx6-1 during mouse islet development[21]. Treatment with an adenovirus encoding Nkx2-2 three days after NPM led to endogenous activation of *NKX6-1* expression with no compromise of fibroblast viability (Fig. 2a). Nkx2-2 also induced *PAX6*, a pan-endocrine gene required to

achieve high levels of islet hormone gene expression during mouse pancreas development[28,29]. Remarkably, ectopic Nkx2-2 reduced NPM-induced *GCG* gene activation without affecting *INS* gene expression (Fig. 2a).

During development, Pax4 and Nkx2-2 are found in β-cell precursors at around the same time, and their parallel activities

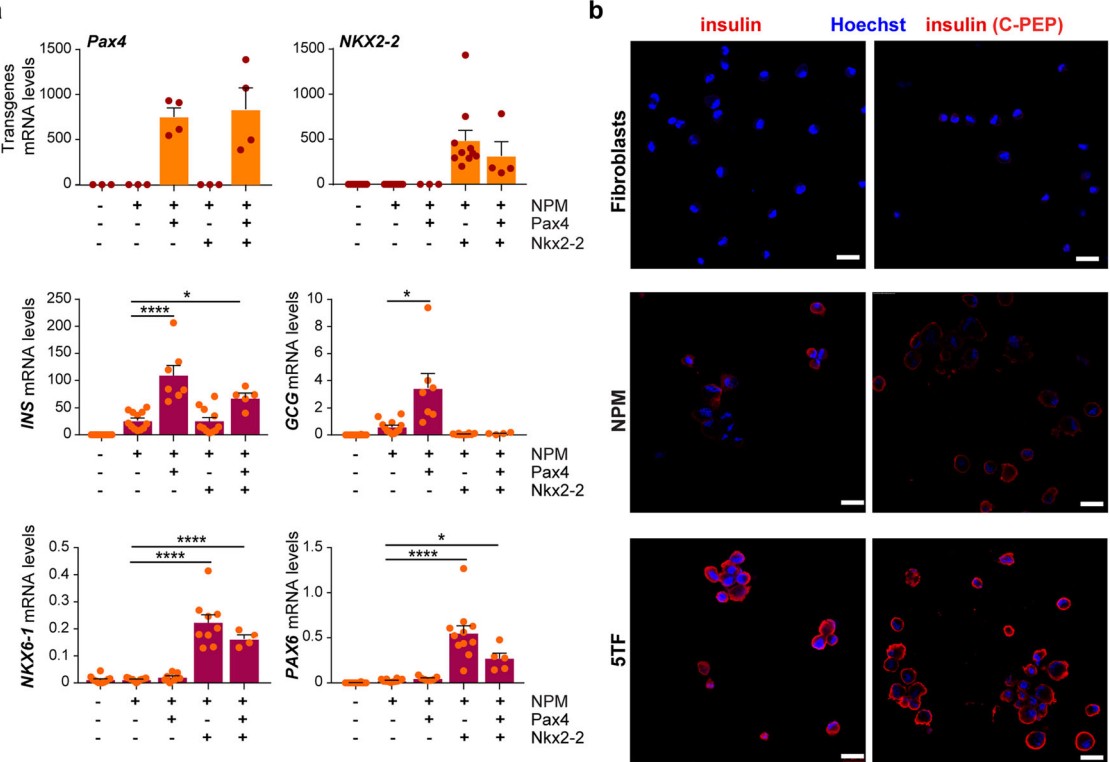

**Fig. 2 Sequential addition of the transcription factors Pax4 and Nkx2-2 enhances β-cell fate in human fibroblasts expressing Neurog3, Pdx1, and MafA.** Human fibroblasts (HFF1) were infected with Ad-NPM alone or sequentially with Ad-NPM and adenoviruses encoding the transcription factors Pax4 and Nkx2-2. Ad-Pax4 and Ad-Nkx2-2 were added three days after NPM in the two-virus conditions. In the three-virus condition, Pax4 was added three days and Nkx2-2 and six days after NPM (condition called 5TF). All cells were collected ten days after infection with Ad-NPM. **a** qPCR of the indicated transgenes and endogenous genes. Expression levels are calculated relative to *TBP*. Values represent the mean ± SEM ($n = 4$–12). **b** Representative immunofluorescence images showing insulin staining (in red) using two different antibodies, one against C-PEP, in untreated fibroblasts and in fibroblasts infected with Ad-NPM alone or with 5TF. Nuclei were stained with Hoechst (in blue). Scale bar, 25 μm. *$P < 0.05$; ****$P < 0.0001$ relative to NPM in (**b**) using one-way ANOVA and Tukey's multiple comparison test.

are thought to enable the β-cell differentiation program[27]. Hence, we tested the effects of including both transcription factors in the reprogramming cocktail. To ensure optimal expression of each transcription factor, we treated cells with Ad-Pax4 and Ad-Nkx2.2 sequentially, at day 3 and day 6 post-NPM, respectively. Following this protocol, the blockade of *GCG* gene activation and the induction of the *NKX6.1* and *PAX6* genes seen with NPM + Nkx2.2, and the higher *INS* expression elicited by NPM + Pax4 relative to NPM alone were all maintained (Fig. 2a). Neither Pax4 nor Nkx2-2, added alone or together, had any impact on the minimal *INS* gene induction shown when the N + P + M factors were delivered via separate adenoviruses to human fibroblasts (Supplementary Fig. 2).

Consistent with the gene expression data, staining for insulin protein was more robust in cells reprogrammed with NPM + Pax4 + Nkx2.2 than in cells reprogrammed with NPM as assessed using two different antibodies, one against human insulin and another against human C-PEP to exclude possible insulin uptake from the media (Fig. 2b). We quantified the immunofluorescence images and found that 67.9 ± 6.2% of cells in the culture were INS + at day 10.

**Characterization of cells generated from fibroblasts using the 5TF- reprogramming cocktail.** From here on, we used the sequential introduction of the five transcription factors (5TF protocol, Fig. 3a) to generate insulin-producing cells from human fibroblasts (reprogrammed cells will be referred as 5TF cells). At day 10, 5TF cells displayed an epithelial morphology (Fig. 3b) and

hadn't grown as much as untreated fibroblasts (day 10; 5TF: $44 \times 10^3 \pm 3 \times 10^3$ cells/well; control: $238 \times 10^3 \pm 18 \times 10^3$ cells/well, $n = 18$). This decreased cell number was likely due to diminished proliferation, which was evident as soon as one day following Ad-NPM infection (Fig. 3c). The capacity of cells to reduce the MTT compound, in contrast, was comparable to that of fibroblasts, indicating that viability was not compromised (Fig. 3d).

Next we studied expression of selected differentiation transcription factor genes at days 10-11 of the protocol. All genes tested, except *PAX4*, were more expressed in 5TF relative to NPM (*NEUROD1, INSM1, HNF1B, MAFB, PDX1, NEUROG3, NKX2.2*) (Fig. 3e). Likewise, several genes (*PCSK1, KCNJ11, GLP1R, NCAM1*) that are linked to β-cell function were increased in 5TF cells as compared to NPM cells (Fig. 3f). Remarkably, some genes were induced *de novo* by 5TF (*ABCC8, GIPR*) (Fig. 3f). In line with a loss of *GCG* activation, the pro-convertase gene *PCSK2*, which is expressed at higher levels in α than in β cells[30], was reduced by 5TF as compared to NPM (Fig. 3f). These results support that sequential introduction of Pax4 and Nkx2-2 after NPM endorses the β-cell differentiation program in human fibroblasts. β-cell gene activation was sustained for at least twenty-one days after initiation of the protocol despite reduced expression of the reprogramming factor transgenes (Fig. 3g, h). Furthermore, expression of several of the tested genes increased with time in culture including *NKX6-1, PCSK1, KCNJ11, ABCC8* and *CHGB* among others (Fig. 3g), suggestive of permanent cell lineage conversion.

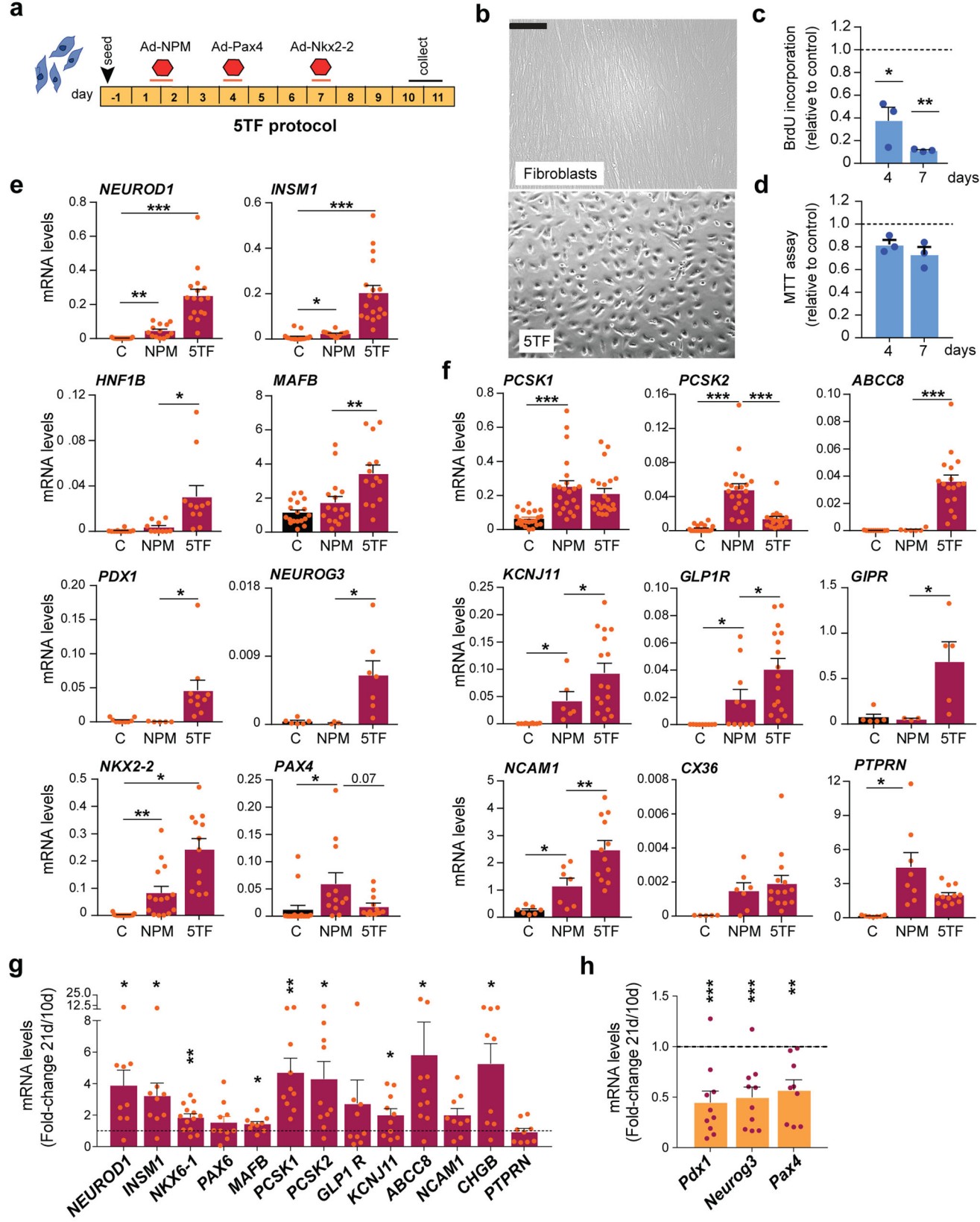

Glucose-induced insulin secretion by β cells is mediated by cellular glucose metabolism, closure of ATP-dependent potassium channels, membrane depolarization and opening of voltage-dependent calcium channels, resulting in an increase in cytosolic $Ca^{2+}$ that triggers insulin exocytosis. We investigated whether 5TF cells increased intracellular $Ca^{2+}$ in response to glucose and membrane depolarization elicited by high potassium. We found that 65% of the cells exhibited a response to glucose, high potassium, or both, whilst 35% of cells were unresponsive to either stimulus (Fig. 4a and Supplementary Video 1). Parental fibroblasts not engineered for 5TF expression were unresponsive to these stimuli (Fig. 4b and Supplementary Video 2). Among

**Fig. 3 The 5TF protocol results in cell growth arrest and activation of endogenous β-cell differentiation transcription factors and β-cell marker genes in human fibroblasts. a** Scheme of the reprogramming protocol 5TF (NPM + Pax4 + Nkx2.2) showing the sequence of addition of adenoviruses encoding the indicated transcription factor/s. Duration of incubation with each adenovirus is represented with a line. Cells were studied at days 10-11 after initial addition of Ad-NPM. **b** Representative bright field image of parental fibroblasts and 5TF reprogrammed fibroblasts at day 10. Scale bar, 75 μm. **c** Cell proliferation measured by BrdU incorporation and **d** cell viability measured by MTT assay for $n = 3$ independent reprogramming experiments. Bars represent values relative to control fibroblasts (given the value of 1, represented by a dotted line). Note that day 4 values are before Pax4 introduction. **e, f** qPCR of islet/β-cell transcription factor and β-cell function genes in untreated control fibroblasts (C, $n = 5$–22), in fibroblasts infected with Ad-NPM alone ($n = 3$–22) or with 5TF ($n = 5$–22). Expression levels were calculated relative to *TBP*. qPCR of the indicated endogenous genes (**g**) and transgenes (**h**) at day 21 after initiation of reprogramming ($n = 9$–13, from 7 reprogramming experiments). Transcript levels are expressed relative to levels in cells at day 10 of the reprogramming protocol (given the value of 1, shown with a dotted line). Data are mean ± SEM for the number of n indicated in parentheses. *$P < 0.05$, **$P < 0.01$, ***$P < 0.001$ compared to control fibroblasts (**c, d**), or between indicated bars using unpaired *t*-test (**e, f**), or compared to day 10 5TF cells (**g, h**) using one-sample *t*-test.

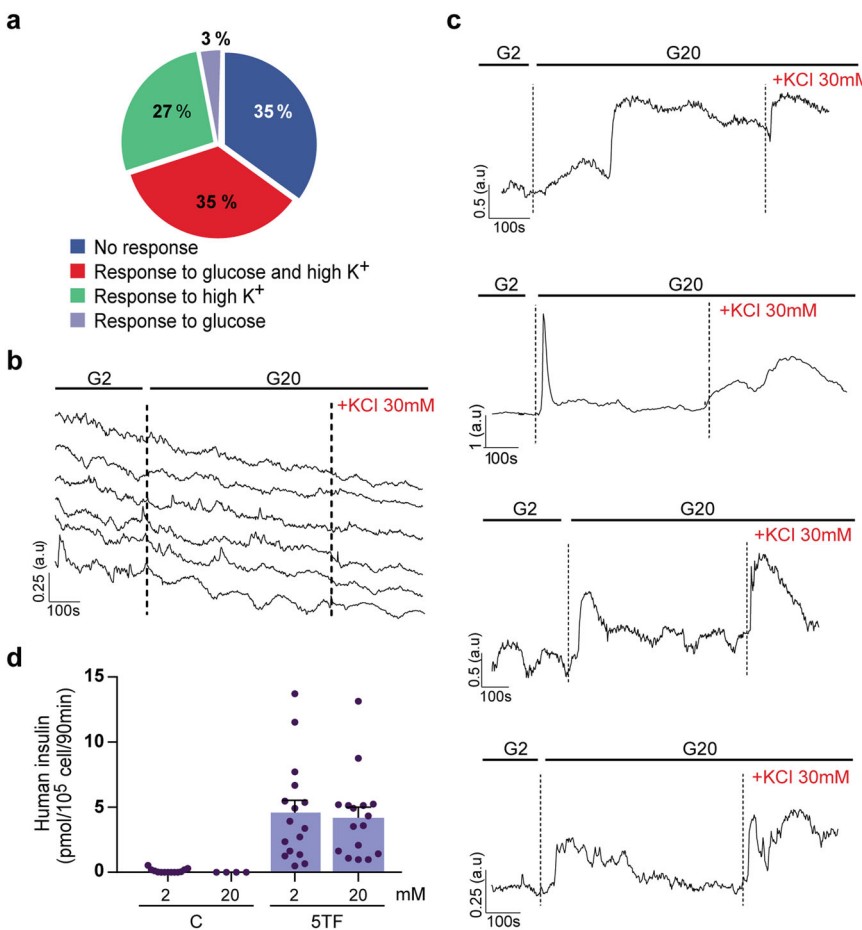

**Fig. 4 5TF cells increase intracellular calcium in response to glucose and KCl.** 5TF cells were loaded with the calcium indicator Fluo-4-AM at day 10 of the reprogramming protocol. Single-cell imaging to detect cytosolic calcium was performed in the following sequence: low glucose (2 mM, G2), high glucose (20 mM, G20) and membrane depolarization with KCl (30 mM). **a** Quantification of the frequency of cells ($n = 200$, from six independent reprogramming experiments) that responded to glucose, membrane depolarization elicited by high potassium or both. Representative measurements of dynamic Fluo-4 fluorescence for (**b**) six fibroblasts and (**c**) four 5TF cells. **d** In vitro insulin secretion by 5TF cells. ELISA determination of secreted human insulin by control fibroblasts ($n = 4$-13) and 5TF cells ($n = 16$) under non-stimulatory conditions (glucose 2 mM) and under stimulatory conditions (glucose 20 mM). Data are mean ± SEM and correspond to six independent reprogramming experiments, 2–4 biological replicates per experiment.

responsive cells, approximately half responded to both glucose and high potassium and half responded only to potassium (Fig. 4a). We observed heterogeneity in the amplitude and kinetics of responses among individual cells (Fig. 4c). Next, we performed static incubation assays to study GSIS and found that 5TF cells released similar amounts of human insulin at low (2 mM) and high (20 mM) glucose concentrations (Fig. 4d). Thus, even though 5TF cells increased their intracellular calcium in response to glucose and membrane depolarization, they secreted insulin in a constitutive manner.

**Generation of 5TF cell spheroids and transcriptome-wide analysis.** The differentiation and functionality of many cell types vary dramatically between three-dimensional (3D) and two-dimensional (2D) monolayer cultures, the former being closer to the natural 3D microenvironment of cells in a living organism. Thus, we generated spheroids of 5TF cells (1200-1800 cells/spheroid; average diameter of $128 \pm 27$ μm) one day after the introduction of Nkx2-2 and maintained them in culture for three additional days (Fig. 5a). At the time of collection, insulin-positive staining was easily identified but glucagon and

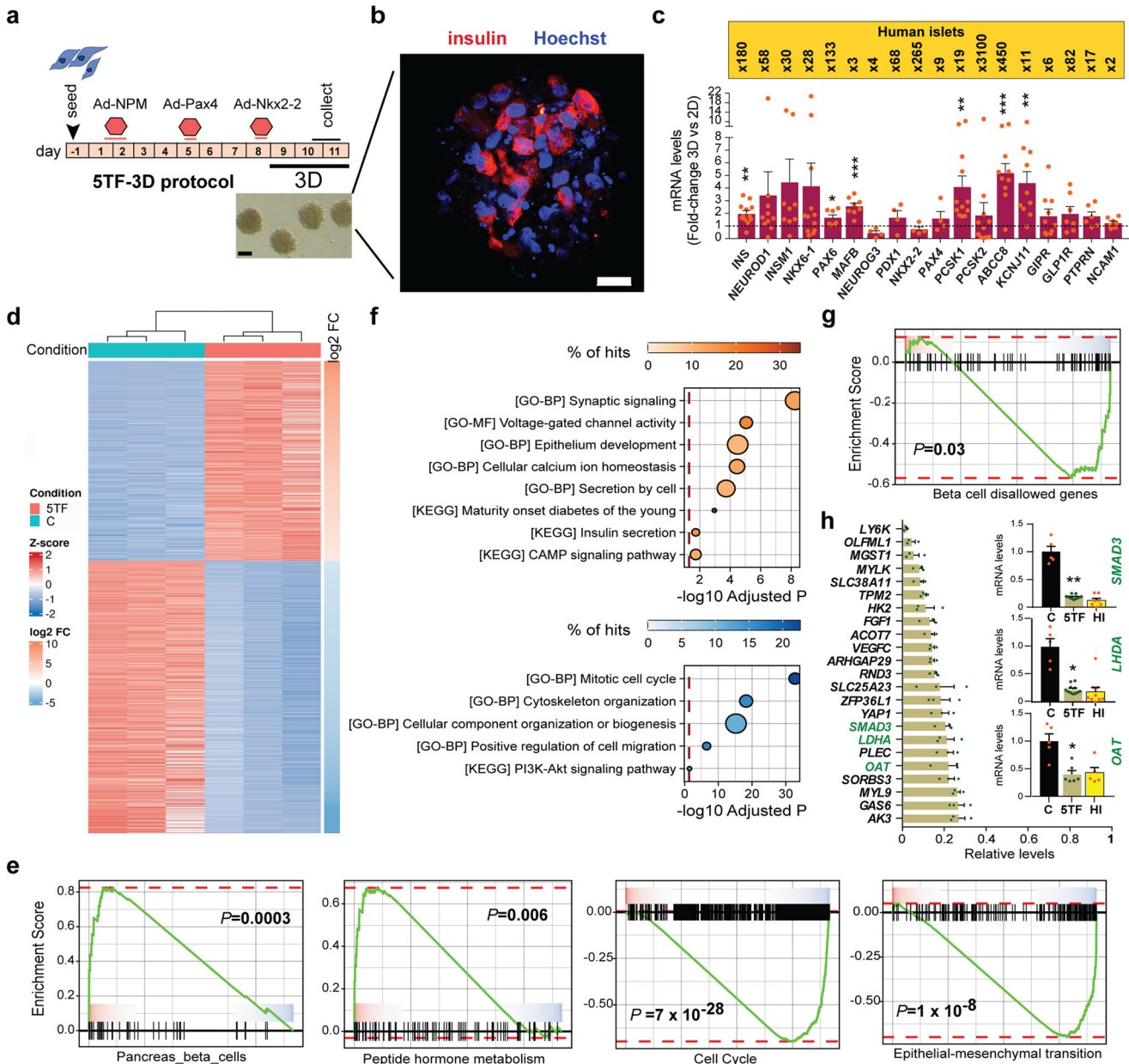

**Fig. 5 Generation and transcriptomic characterization of 5TF cell spheroids. a** Schematic representation of the modified 5TF protocol (5TF-3D): cells were moved from 2D to 3D culture during the last three days (days 7–10) of the protocol. Representative bright field image of 5TF cell spheroids. Scale bar, 100 μm. **b** Representative immunofluorescence image showing insulin staining in red and nuclei in blue (marked with Hoechst) of a 5TF cell spheroid at the end of the reprogramming protocol. Scale bar is 50 μm. **c** qPCR of the indicated genes in 5TF cell spheroids. Transcript levels are expressed as fold relative to levels in 5TF cells maintained in 2D culture throughout the 10-day protocol (given the value of 1, dotted line). Data are mean ± SEM for $n = 4–12$. *$P < 0.05$, **$P < 0.01$, ***$P < 0.001$ relative to 2D culture using one-sample $t$-test. Fold-change differences in expression levels between human islets and 3D-5TF reprogrammed cells are shown in the upper yellow box. **d** Heat map of differentially expressed genes between parental fibroblasts (C) and 5TF cell spheroids ($n = 3$ reprogramming experiments). **e** GSEA plots on indicated gene sets and pathways. **f** Dot plots showing the enrichment analysis on Gene Ontology (GO) and KEGG categories of differentially expressed genes (gained in red, lost in blue) between fibroblasts (C) and 5TF cells. The $X$-axis represents the adjusted $p$ value, the size of the dot represents the number of enriched genes (count) and the color intensity of the dots represents the percentage of hits in each category. **g** GSEA plot on β-cell disallowed genes. **h** Relative expression levels of β-cell disallowed genes repressed in 5TF cell spheroids as compared to fibroblasts (given the value of 1) based on RNA-seq data normalized expression values. Data are mean ± SEM ($n = 3$). Insets show mRNA expression of the indicated genes in untreated control fibroblasts ($n = 5$), 5TF cell spheroids ($n = 6$) and human islets ($n = 5$) as assessed by qPCR. Expression levels were calculated relative to *TBP*. Data are mean ± SEM. *$P < 0.05$, **$P < 0.01$ relative to control fibroblasts using unpaired $t$-test.

somatostatin staining was undetectable (Fig. 5b and Supplementary Fig. 3). While *INS* transcript levels were nearly 2-fold higher in 5TF cell spheroids compared to 5TF cells kept in monolayer, other β-cell marker genes, such as the prohormone convertase *PCSK1* and the ATP-sensitive potassium channel

subunits *KCNJ11* and *ABCC8*, showed a higher response (4 to 5-fold) to 3D culture (Fig. 5c). Thus, cell aggregation during the last stage of reprogramming (note that total length of the protocol was not changed) conferred improved activation of genes associated to β-cell function. Despite increased gene activation, β-cell

gene expression in 5TF cell spheroids remained lower than in human islets, with differences ranging widely among examined genes (Fig. 5c).

To obtain a more comprehensive understanding of the cell identity switch induced by the 5TF-3D reprogramming protocol, we performed RNA-sequencing of 5TF cell spheroids and parental fibroblasts. A total of 2806 genes (1186 upregulated, 1620 downregulated) were differentially expressed between both cell populations (adjusted p-value <0.05 and fold-change (FC) > 2) (Fig. 5d and Supplementary Data 1). Gene set enrichment analysis (GSEA) showed that pancreas/β-cell and peptide hormone metabolism gene sets were enriched in 5TF cells (Fig. 5e). Biological functions associated with gained genes included epithelium development, synaptic signaling, ion transport, calcium sensing and secretion (Fig. 5f). Among the upregulated genes related to stimulus-secretion coupling, there were synaptotagmins (*SYT1,2,3,6,13,17*), syntaxins (*SYN2 SYN3*), calcium sensors (*SCG2*) and SNARE protein complexes (*VAMP1)*. Correlating with our previous results, cell cycle and mitotic function genes were enriched among repressed genes (Fig. 5e, f). Additionally, GSEA demonstrated that 5TF cells had a lower expression of the gene set associated with the epithelial-mesenchymal transition (Fig. 5e). In agreement, functions including cytoskeleton organization and cellular migration were overrepresented among lost genes (Fig. 5f). Interestingly, GSEA also revealed that the β-cell disallowed gene set, which includes genes that are selectively suppressed in β cells and believed to be detrimental for β cell function[31–33], was reduced in 5TF cells (Fig. 5g). A total of 23 previously recognized β-cell disallowed were significantly downregulated in 5TF cells (Fig. 5h). By using qPCR, we confirmed the repression of three of these genes -*OAT, LDHA,* and *SMAD3*- which are regarded as part of the core disallowed unit[33]. Of note, the levels of these genes in 5TF cells matched those of human islets (Fig. 5h). Collectively, these results show that 5TF-3D reprogramming promotes a change in the fibroblast transcriptome, including selective gene activation along with specific gene repression events, enabling a change in cell identity from fibroblast towards a β-cell fate.

**Ultrastructure and insulin secretory features of 5TF cell spheroids**. Consistent with gene activation events identified in prior gene expression analyses, immunofluorescence staining showed the presence of the mature β-cell markers PCSK1, NCAM1, and KCNJ11 (Kir6.2) in many insulin-positive 5TF cells. PTPRN (IA2) was also expressed albeit more sporadically in insulin-positive 5TF cells (Fig. 6a). Using conventional electron microscopy, we looked for the existence of secretory granules and discovered that most cells contained multiple spherical electron-dense prototypical secretory vesicles (Fig. 6b). These vesicles showed a high degree of morphological heterogeneity, presumably as consequence of their degree of maturation and/or loading. Although they did not have the appearance of typical insulin-containing granules from primary β cells, which are characterized by a clear halo surrounding a dark polygonal dense core[34], some of the vesicles exhibited a gray or less electron dense halo and looked like the granules described in immature insulin-positive cells generated in early stem cell differentiation protocols[35,36].

We next performed static incubation GSIS assays. 5TF cell spheroids exhibited significant insulin secretory response to glucose (fold 20 mM/2 mM: 2.02 ± 0.18) as compared to 2D cultures (fold 20 mM/2 mM: 1.08 ± 0.15) (Fig. 6c, d). To establish the glucose threshold for stimulation of insulin secretion, 5TF cell spheroids were subjected to either 2, 5, 11 or 20 mM glucose. Between 2 mM and 11 mM/20 mM glucose, 5TF spheroids showed a 2.3-fold increase in insulin production on average

(Fig. 6e). In contrast, although there was some variability, they did not show a statistically significant increase in insulin secretion between 2 mM and 5 mM glucose (Fig. 6e). These observations indicate that 5TF cell spheroids are stimulated at higher glucose threshold; it is interesting to note that human islets have a glucose threshold at 3 mM and a maximal response at 15 mM[37].

The 5TF-3D protocol was repeated on an additional HFF line and produced results that were comparable (Supplementary Fig. 4) proving the reproducibility of the reprogramming protocol.

**Transplantation of 5TF cell spheroids**. Finally, we studied the stability of reprogramming in vivo. With this aim, we transplanted 300 5TF cell spheroids (1000–1200 cells/spheroid) into the anterior chamber of the eye (ACE) of non-diabetic immune-deficient NOD *scid* gamma (NSG) mice (Fig. 7a). The ACE allows fast engraftment[38] and in vivo imaging[39]. Ten days following transplantation, we used two-photon microscopy to evaluate in vivo graft re-vascularization and confirmed the presence of functioning vessels in the grafts (Fig. 7b). Additionally, by observing the long-term tracer CFDA's fluorescence, we confirmed that the transplanted cells were alive (Fig. 7b). To assess the maintenance of insulin expression in vivo, we harvested the eye grafts at day 10 for RNA extraction and immunostaining. Human *INS* mRNA was readily detectable and levels, calculated relative to human *TBP*, were comparable to those in 5TF cell clusters prior to transplantation (Fig. 7c). In agreement, abundant HLA + (human cell marker) cells that stained for insulin were detected in the eye grafts by immunofluorescence staining (43.5 ± 2.8% INS+HLA+/total HLA+, *n* = 5) (Fig. 7d, e and Supplementary Fig. 5). We observed positive staining for the reprogramming transcription factors in 20–30% of the INS + cells (Supplementary Figure 6). Although we were unable to discriminate between the two, high transgene expression found by qPCR analysis in eye grafts (Supplementary Fig. 6) indicated that the staining represented virally encoded exogenous protein rather than endogenous protein. Since adenoviral vectors do not normally integrate into the host DNA, we speculate that the cessation of cell division induced by reprogramming may explain persistent transgene expression in 5TF cells. In fact, similar findings were reported in reprogrammed human duct-derived insulin-producing cells[9]. We were able to identify INS + cells in 4 (of 5) grafts harvested one month after transplantation even though their number was reduced relative to day 10 grafts (Supplementary Fig. 7). The proportion of INS+HLA+ cells in 30-day grafts was more heterogeneous than in 10-day grafts, and in 3 (of 5) grafts, it was comparable or even higher than that of 10-day grafts, demonstrating the maintenance of reprogramming (Supplementary Fig. 7).

To study if 5TF cells secreted insulin in vivo, we first measured the presence of human insulin by ELISA in the aqueous humor of the transplanted eyes. Human insulin was readily detectable in eyes carrying 5TF cell grafts (17 of 17, ranging from 76 to 1103 pmol/L) whilst no insulin was detected in eyes transplanted with parental fibroblast clusters or in non-transplanted mice (Fig. 7f). For comparison, eyes containing 300 5TF spheroids showed on average approximately 20-fold lower levels of human insulin than eyes containing 150–200 human islets (Fig. 7f). Due to space limitations in the ACE, we transplanted a larger number of spheroids (3500–5000) into the omentum of normoglycemic NSG mice in order to detect circulating human insulin in host animals. We measured low amounts of human insulin in the plasma of most transplanted mice, and these levels increased in 6 (of 10) mice after receiving an intraperitoneal glucose injection on day 30 post-transplantation (3.6 ± 0.9 vs 13.9 ± 3.7pmol/L, *p* = 0.014)

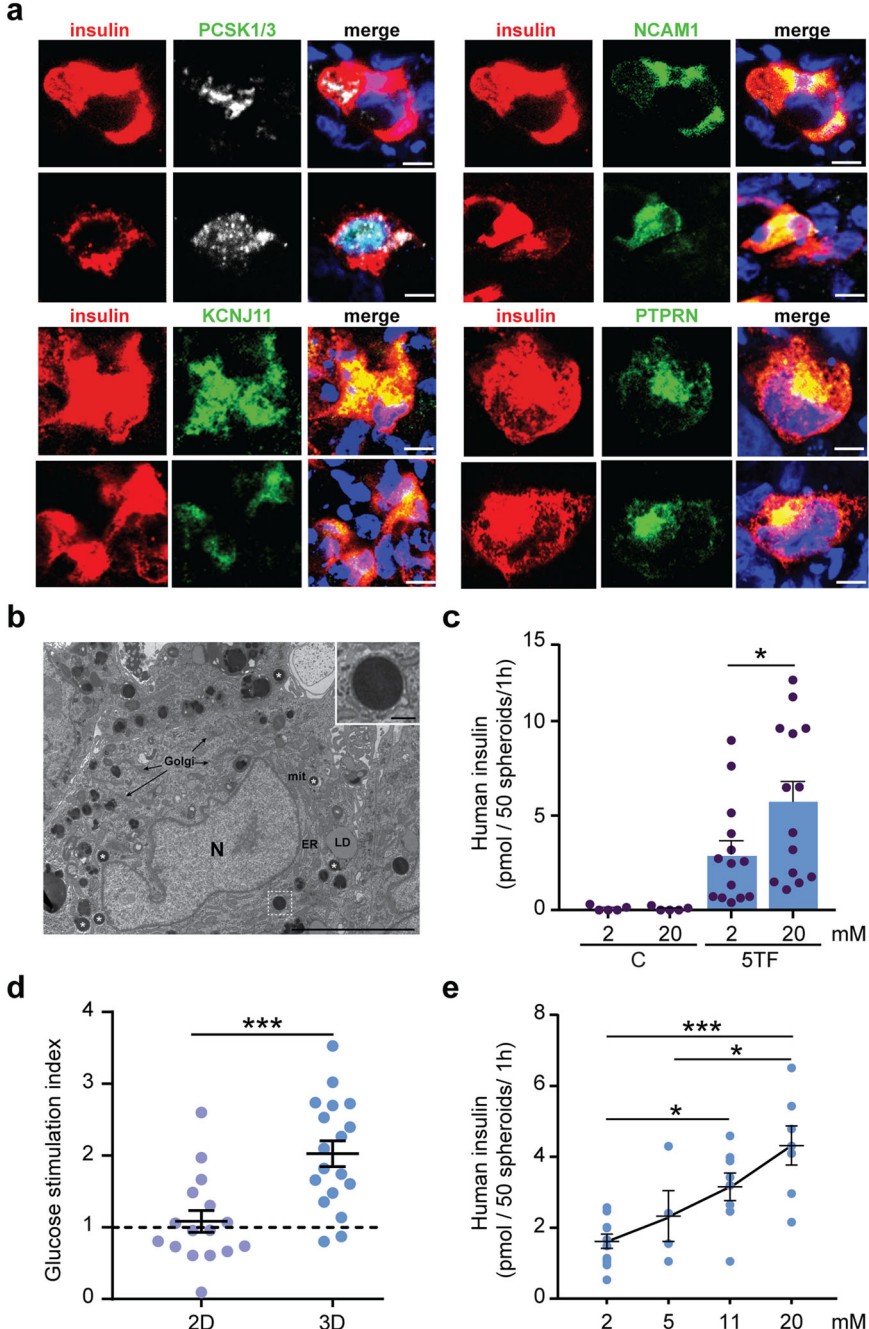

**Fig. 6 Insulin secretion by 5TF cell spheroids. a** Representative confocal images of 5TF cell spheroids immunostained with the indicated antibodies. Scale bar, 10 μm. **b** Conventional transmission electron microscopy showing a representative image of a 5TF cell spheroid. Prototypical electron dense secretory vesicles (asterisks) are observed dispersed in the cytoplasm. Well-preserved mitochondria (mit), endoplasmic reticulum (ER), Golgi membranes (G) and lipid droplets (LD) are also observed. Inset shows a detail of a secretory vesicle with an average diameter of 450 nm. N, nucleus. Scale bars are 200 nm (inset) and 500 nm. **c** In vitro glucose-induced insulin secretion by 5TF cell spheroids ($n = 14$, from 8 reprogramming experiments). Secretion by control spheroids composed of parental fibroblasts ($n = 5$) is also shown. **d** Glucose stimulation Index (ratio between insulin secreted at 20 mM glucose vs. 2 mM glucose) of 5TF cells maintained in 2D or in 3D (spheroid) cultures ($n = 16$–18, from 8 to 10 reprogramming experiments). **e** Glucose dose curve of insulin secretion by 5TF cell spheroids ($n = 4$–12, 5 reprogramming experiments). Data are presented as the mean ± SEM for the number of n indicated in parentheses. *$P < 0.05$; ***$P < 0.001$ between the indicated conditions using unpaired $t$-test (**c**), one sample $t$-test (**d**) or one-way ANOVA (**e**).

(Supplementary Fig. 8). Transplants were repeated in other locations yielding similar results (Supplementary Table 2). As observed in the ACE grafts, a low number of INS + cells were identified in omentum grafts harvested at 30 days post-transplantation (Supplementary Fig. 8). These findings show that, despite restricted survival, reprogramming is maintained

and 5TF cells maintain the capacity to release insulin in an in vivo setting.

## Discussion

This study describes a direct reprogramming protocol based on the sequential introduction of five lineage-determining

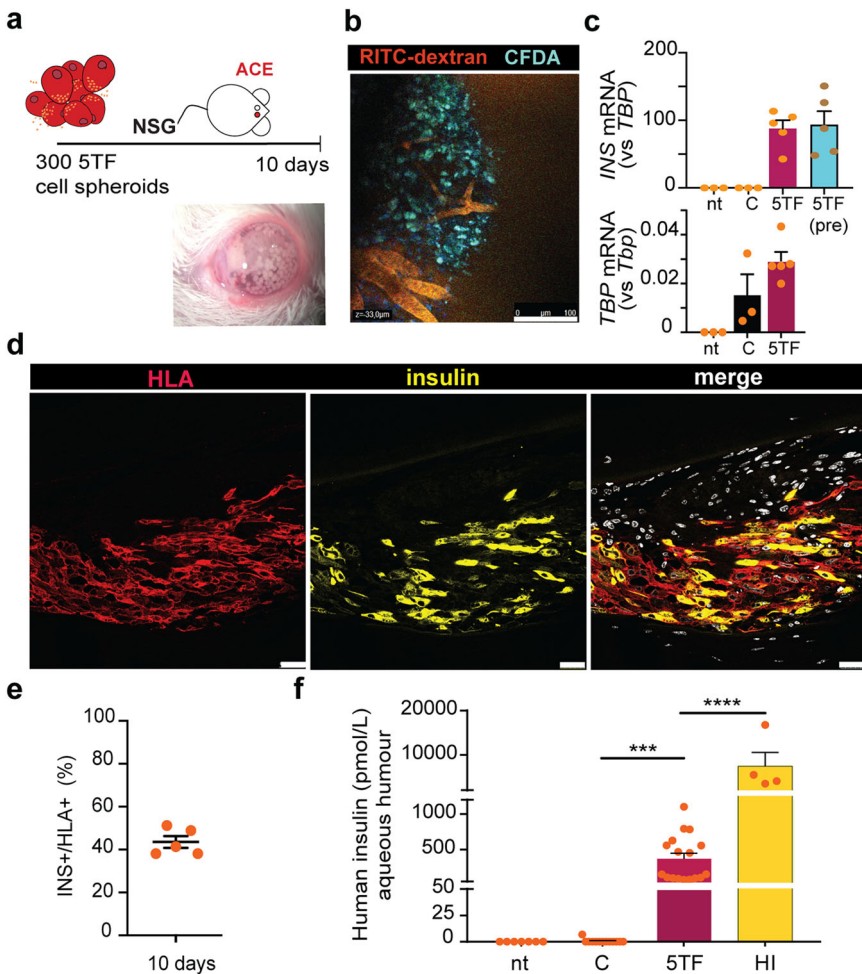

**Fig. 7 In vivo characterization of 5TF cell spheroids. a** Schematic illustration and image showing 5TF cell spheroids transplanted into the anterior chamber of the eye (ACE) of a normoglycemic NSG mouse. **b** Vascularization of 5TF cell grafts ten days following transplantation into the ACE. Representative in vivo image depicting functional vessels (RITC-dextran, red) and viable 5TF cells (CFDA, green). Scale bar, 100 μm. **c** qPCR of *INS* and *TBP* transcripts in eyes of non-transplanted mice (nt, *n* = 3) and mice transplanted with either control fibroblast spheroids (C, *n* = 3) or 5TF cell spheroids (*n* = 5) collected ten days post-transplantation. *INS* gene expression in 5TF cell spheroids prior to transplantation is depicted in the blue bar (*n* = 6). *INS* gene expression is calculated relative to *TBP*. Expression of *TBP* relative to mouse *Tbp* is shown to prove the presence of human cells in eyes receiving control and 5TF spheroids. Data are presented as mean ± SEM. **d** Representative immunofluorescence images showing HLA staining in red and insulin staining in green in 5TF cell grafts ten days post-transplantation. Scale bar, 25 μm. **e** Percentage of cells doubly positive for insulin and HLA (relative to total HLA + cells) in 5TF cell grafts at day 10 following transplantation. Each dot corresponds to one eye graft (*n* = 5). **f** ELISA determination of human insulin in the aqueous humor in un-transplanted mice (*n* = 7), in mice transplanted with either 300 fibroblast spheroids (*n* = 14) or 300 5TF cell spheroids (*n* = 17) at day 10 post-transplantation and in mice transplanted with 150–200 human islets (*n* = 4) at day 12–15 post-transplantation. Data are presented as mean ± SEM for the number of n indicated in parentheses. ***P < 0.001; ***P < 0.0001 between indicated samples using unpaired *t*-test.

transcription factors that induces β-cell fate in human fibroblasts. Reprogrammed fibroblasts exhibit the concomitant activation of β-cell genes and the repression of fibroblastic and β-cell disallowed genes. Significantly, reprogrammed cells display functional features of β cells, including the ability to mobilize calcium and secrete insulin upon glucose stimulation. To the best of our knowledge, this is the first instance where it has been shown that skin fibroblasts can serve as cells of origin for β-cell derivation using transcription factor-based direct conversion methodologies.

The N + P + M cocktail was initially described for the conversion of pancreatic exocrine cells into β cells in the mouse pancreas in situ[4]. Following studies demonstrated that these factors were also able to reprogram other endoderm-derived gastrointestinal tract cell lineages towards β-like cells[6–9]. Contrarily, information in the literature suggested that these transcription factors were ineffective in promoting β-cell fate from mesoderm-derived mouse and human fibroblasts[4,9,18]. However,

in our view, the available studies did not fully analyze this option. In the present work, we examined a range of simultaneous and sequential transcription factor combinations, and were able to show that the N + P + M cocktail also works in human fibroblasts. Successful reprogramming is known to depend on a number of variables in addition to the selection of the proper reprogramming cocktail. We found that reprogramming was only possible when the three transcription factors where supplied by a single polycistronic adenovirus rather than multiple adenoviruses that expressed them separately, which is line with earlier research employing this same vector in acinar cells[19]. This may be owing to the random nature of viral co-infection, which makes it impossible to ensure that every infected cell receives every transgene when employing distinct viruses. Two other crucial parameters in reprogramming are the stoichiometry and the expression level of the reprogramming factors[2,3,40,41]. Regarding the former, a polycistronic expression system instead of using

several vectors offers a more homogenous TF stoichiometry across the infected cells. We speculate that successful reprogramming in our study was made possible by the use of this polycistronic construct and the high levels of transgene expression achieved in fibroblasts through a minor modification of the infection protocol. It is interesting that the NPM factors, identified as a β-cell promoting reprogramming cocktail in acinar cells, induced expression of the *GCG* and *SST* genes in human fibroblasts. Despite the fact that *INS* was the most activated by difference, we used other factors (Pax4, Nkx2-2) to confer enhanced β-cell specificity in fibroblasts. These observations highlight the need to customize conversion transcription factor cocktails to the selected cell source.

The 5TF-3D protocol used in our study promoted transcriptome alterations consistent with a change in cell identity from fibroblast toward a β-like cell. It is noteworthy that nearly 60% of the detected transcriptional changes are gene repression events. In addition to the downregulation of fibroblast-specific genes, we found that 5TF cells had selectively suppressed the group of β-cell disallowed genes. This finding suggests that, in a cellular environment of a different lineage, lineage-specific transcription factors can promote the precise suppression of potentially damaging genes for their lineage. Intriguingly, *SLC16A1* (MCT1), one of the founding members of the β-cell disallowed gene set, was not repressed in 5TF cells. Developmentally, β-cell disallowed genes are marked for repression by Polycomb Group (PcG) proteins in pancreatic progenitors during pancreas organogenesis[42,43]. However, the mechanisms that support their silencing in adult cells are still poorly understood and might vary amongst genes. These results spur future experiments into the way developmental transcription factors repress undesirable genes during the reprogramming process.

Experimental data showing that intracellular calcium concentration and insulin release were stimulated by glucose suggest that fibroblast-derived 5TF cells are progressing toward a β-cell at the functional level. The levels of *INS* gene activation reached in 5TF cells were higher than those of reprogrammed human hepatocytes[44] and comparable to those reported for reprogrammed human pancreatic duct cells[9]. They were, nonetheless, less than those of human islets. 5TF cells share traits with first generation pluripotent-derived β-like cells, such as decreased insulin production, low β-cell gene expression, and the absence of mature insulin granules[36,45]. Hence, we acknowledge that there is room for advancement and that optimization of our current reprogramming procedure is required in order to generate cells that are more similar to a primary β cell. A potential strategy is to modify the cell culture settings. Here, for instance, we demonstrated how switching to a different culture media after the addition of the NPM factors noticeably improved *INS* gene activation. We also showed how switching 5TF cells from a 2D to a 3D culture system increased expression of key β-cell genes and enabled converted cells to develop the ability to release insulin in response to glucose. Thus, additional adjustments to culture conditions, like extended culture times or agitation, may be helpful to improve quality of the generated cells[46]. In this same line, inclusion of soluble signaling molecules during or after introduction of the conversion transcription factors could be employed[47–49], as it has been shown in direct reprogramming examples towards neurons[50]. In recent years, stem cell research has developed a significant body of knowledge on how to improve the maturity of pluripotent cell-derived insulin-secreting cells created in vitro[22,23,35,36,51–56]. We anticipate that this information can be very helpful in the effort to optimize direct reprogramming approaches from somatic cell types toward the β-cell lineage.

Poor long-term survival of 5TF cells after transplantation precluded physiological studies in mouse models. Although the causes of cell loss remain to be elucidated, limited graft survival is a prevalent concern in transplantation of cadaveric donor islets and in vitro created islet tissue[57]. In our case, it is possible that additional endocrine and/or non-endocrine cell types normally found in islets such as endothelial cells are required for better engraftment and prolonged survival[58,59]. Additionally, our observation of sustained expression of exogenous reprogramming factors in 5TF cells raises the question of what effects this might have, especially the continued presence of Neurog3 and Pax4 given that these factors are not present in mature β cells[27,60]. Future effort will be required to develop reprogramming strategies to guarantee that the conversion factors are turned off if necessary.

The generation of substitute β cells from a cell source that can be replenished has been a long-standing major goal in diabetes research. The most advanced approach to date involves the guided differentiation of pluripotent stem cells to islet cells. The first clinical studies involving actual patients have been made possible by the enormous advancements made in this field over the past fifteen years[36,45,46,54,56,61,62]. Recent publication of the first mid-term results of one of these trials shows positive outcomes as well as the need to continue improving current differentiation protocols and transplantation techniques[63,64]. On the other hand, the idea of direct cellular reprogramming to produce therapeutically relevant cell types, as an alternative to their derivation from stem cells, regained momentum with the discovery of iPS cells[65,66]. Since then, there has been a notable increase in the amount of somatic cells produced from readily accessible cell types, such as fibroblasts[67–69]. In the β-cell field, however, available evidence indicated that human fibroblasts were not keen to change identity toward a β cell via direct reprogramming. Our results challenge this view and show that this is possible if the appropriate combination of conversion factors and conditions are found. The value of our approach is related to the major benefit of using a cell source that is readily available, such as the skin fibroblast, in terms of translational potential. Two further assets that should be taken into account from a clinical standpoint are the possibility of auto-transplantation and the avoidance of tumor-related concerns linked to pluripotent cell states. Finally, the relative simplicity of direct reprogramming methodology compared to pluripotent stem cell derivation supports the ongoing interest in developing this kind of approaches.

In conclusion, here we demonstrate that human fibroblasts can be directly converted toward a β-cell fate using a defined set of developmental transcription factors. Further research should refine this strategy so that generated insulin-producing cells more closely resemble primary β cells. These findings provide a promising starting point for future investigation into an alternative pathway to produce β-like cells for therapeutic and modeling purposes.

## Methods
**Fibroblasts**. Human fibroblasts were obtained from a child foreskin biopsy after signed informed content and approval of the institutional Review Board of the Center of Regenerative Medicine in Barcelona. In brief, skin biopsy was collected in sterile saline solution, divided into small pieces, and allowed to attach to cell culture dishes before adding Iscove's modified Dulbecco's medium (Invitrogen, Carlsbad, CA, USA) supplemented with 10% human serum (Sigma, St. Louis, MO, USA) and penicillin/streptomycin (0.5X) (Invitrogen). After 10 days of culture at 37 °C, 5% $CO_2$, fibroblast outgrowths were dissociated and split 1:4 using a recombinant trypsin-like enzyme (TrypLE Select, Invitrogen). Preparation was negative for hematopoietic markers, including CD34. This fibroblast line (HFF1) was used to design the reprogramming protocol. Once established, the protocol was validated in another HFF preparation (HFF2) that was purchased from a commercial source (SCRC1041^TM, ATCC, Manassas, VA, USA).

**Human islets**. Human islets were prepared by collagenase digestion followed by density gradient purification at the Laboratory of Cell Therapy for Diabetes (Hospital Saint-Eloi, Montpellier, France), as previously described[70]. After reception in Barcelona, human islets were maintained in culture at 37 °C, 5% $CO_2$ for 1–3 days in RPMI-1640 with 5.5 mM glucose, 10% fetal bovine serum (FBS) and antibiotics, before performing the experiments. Experiments were performed in agreement with the local ethic committee (CHU, Montpellier) and the institutional ethical committee of the French Agence de la Biomédecine (DC Nos. 2014-2473 and 2016-2716). Informed consent was obtained for all donors.

**Recombinant adenoviruses**. The adenoviral expression vector pAd/CMV/V5-DEST carrying mouse Neurog3, Pdx1, MafA and 2A-Cherry under the CMV promoter and separated by self-cleaving 2A peptides was kindly provided by Dr. Q. Zhou, Cornell University[19]. The recombinant adenovirus (hereafter termed Ad-NPM) was generated after Pac1 digestion and transfection into HEK293 cells. The recombinant adenovirus encoding Pdx1 was kindly provided by the Beta Cell Biology Consortium. The recombinant adenovirus encoding MafA was purchased from Vector Biolabs (Chicago, IL, USA). All other recombinant adenoviruses encoding single transcription factors (Neurog3, NKX2-2, Nkx6.1, and Pax4) were described previously[71,72]. Crude virus lysates were used for infection of fibroblasts.

**Reprogramming protocol**. Fibroblasts were grown in DMEM-F12 media supplemented with 10% (v/v) fetal bovine serum (FBS), 100 U/ml penicillin, 100 µg/ml streptomycin, and 1% Glutamax. They were plated onto 96-well plates (9500 cells per well) for MTT and BrdU assays, onto 12-well plates ($1.25 \times 10^5$ cells/well) for gene expression, insulin secretion, immunofluorescence and caspase assays and onto 10 cm plates ($1.5 \times 10^6$ cells) or T-75 flasks ($3.0 \times 10^6$ cells) for transplantation experiments. Reprogramming was initiated when fibroblasts reached 80% confluence, normally 1–2 days post seeding. Cells were sequentially incubated with 15 moi (multiplicity of infection) of Ad-NPM (day 1), 50 moi of Ad-Pax4 (day 4), and 50 moi of Ad-Nkx2-2 (day 7). As fibroblasts show limited infection by adenoviral vectors[73,74], we added a DNA transfection reagent to the virus incubation conditions to improve transgene expression. This small change significantly increased viral transduction efficiency and allowed human fibroblasts to express significant amounts of the reprogramming factors (Supplementary Fig. 9). This reagent was Superfect (Qiagen, Venlo, Netherlands) or jet-PEI (Polyplus, Illkirch, France). In brief, for one well of a 12-well plate, the appropriate amount of virus was pre-incubated with 1.5–2.0 µl of Superfect or JetPEI in 0.1 ml un-supplemented media for 10–15 min at room temperature. Then, the virus/transfection reagent mix was gently mixed with 0.9 ml supplemented media and added to the cells. Cells were incubated with the virus for 16–18 h (NPM) or 6–8 h (Pax4/Nkx2-2). The volume of culture media, adenovirus crude lysate, and transfection reagent used was scaled down or up according to the well size. After NPM virus removal, media was changed to RPMI-1640 medium containing 6% FBS and antibiotics, and this media formulation was maintained throughout the remaining of the reprogramming protocol. Adenovirus doses and/or time of addition were picked in pilot experiments based on a compromise between high *INS* gene expression and low cytotoxicity.

Spheroidal cell aggregates (spheroids) were prepared one day after Ad-Nkx2.2 infection. Cells were trypsinized and transferred to 96-well Nunclon Sphera plates (Thermo Scientific) to generate spheroids of 1200–1800 cells each, which were used for gene expression, immunofluorescence and insulin secretion assays. For transplantation purposes, spheroids (1000–1200 cells/spheroid) were generated in AggreWell-400 plates (StemCell Technologies, Saint Égrève, France). Cell aggregation was performed in RPMI-1640 media supplemented as detailed above.

**Gene expression assays**. Total RNA from cultured cells was isolated using NucleoSpin®RNA (Macherey-Nagel Düren, Germany) following the manufacturer's manual. Total RNA from eyes was extracted using Trizol reagent (Sigma) and then cleaned and DNAse-treated using RNeasy mini columns (Qiagen) prior to cDNA synthesis. First-strand cDNA was prepared using Superscript III Reverse Transcriptase (Invitrogen) and random hexamers in a total volume of 20 ul and 1/40 to 1/200 of the resulting cDNA was used as a template for real time PCR reactions. Real time PCR was performed on an ABI Prism 7900 detection system using Gotaq master mix (Promega, Madison, WI, USA). Expression relative to the housekeeping gene *TBP* was calculated using the delta(d)Ct method and expressed as 2^(-dCT) unless otherwise indicated. Primer sequences are provided in Supplementary Table 1.

**Proliferation and viability assays**. For quantification of cell proliferation, cells in 96-well plates were cultured overnight with medium containing 5-bromo-2'-deoxuridine (BrdU). BrdU incorporation was determined colorimetrically with the Cell proliferation ELISA kit (Roche, Basilea, Switzerland) following the manufacturer's instructions. For assessing cell viability, cells grown in 96-well plates were incubated with medium containing 0.75 mg/ml of 3-(4,5-dimethythiazol-2-yl)-2,5-diphenyltetrazolium bromide (MTT) for 3 h at 37 °C. The resulting formazan crystals were solubilized in Isopropanol/0.04 N HCl solution and optical density was read at 575 and 650 nm using a Synergy HT reader (BIO-TEK Instruments, Winooski, VT, USA). The OD (575-650) was expressed relative to control fibroblasts, which were given the value of 100%.

**Calcium imaging**. To study glucose-dependent calcium influx, cells were washed with Hanks' Balanced Salt Solution (HBSS, Sigma) and incubated with Fluo4-AM (Life Technologies) in fresh Hepes-buffered Krebs-Ringer buffer (Krb) containing 2 mM glucose for 1 h at 37 °C in the dark. Intracellular calcium fluorescence was recorded from fluo-4-loaded cells using a Leica TCS SPE confocal microscope with an incubation chamber set at 37 °C, and a 40× oil immersion objective. Fluorescent images and average fluorescence intensity were acquired at 600 Hz every 1.8 s, using a 488 nm excitation laser, an emission set at 520 with a bandwidth of 10 nm. Image registry consisted of: 5 min in 2 mM glucose-Krb buffer, 10 min in 22 mM glucose-Krb buffer and 5 min in 30 mM KCl-Krb buffer. Average fluorescence intensity images of each individual cell were analyzed with LAS AF Lite and Fuji programs.

**Insulin secretion assays**. Cells in 2D or 3D aggregates were washed with phosphate-buffered saline (PBS) and then incubated in Krb for 45 min at 37 °C with low glucose (2 mM) to remove residual insulin. Cells were incubated in Krb with low glucose for 90 min and supernatant collected. Then clusters were incubated in Krb with high glucose (20 mM or as indicated in figure) for 90 min and supernatant collected. Human insulin in the collected supernatants was measured using a human Insulin ELISA kit (Crystal Chem, Zaandam, Netherlands).

**RNA sequencing**. Total RNA was extracted using NucleoSpin®RNA (Macherey-Nagel). Quantity and quality of the RNA was assessed using Qubit fluorimeter and TapeStation Instrument, respectively. All samples used for sequencing had RIN > 9. mRNA strand-specific RNA libraries were generated using 200 ng of total RNA with the Illumina® Stranded mRNA Prep Ligation kit and IDT for Illumina RNA UD Indexes following the manufacturer's instructions. Each library was sequenced on an Illumina NextSeq2000 (Illumina, Inc.) in paired-end mode with a read length of $2 \times 50$ base pair. More than 60 million paired-end reads were generated for each sample/condition. Quality of sequenced reads was assessed using FastQC (http://www.bioinformatics.babraham.ac.uk/projects/fastqc/). To better discern between reads coming from mouse transgenes and from human endogenous genes, we generated a custom transcriptome by adding the Gencode M10 mouse transcripts from the transgenes *Pdx1*, *Pax4*, *Neurog3* and *Mafa* to the human GENCODE release 39 human transcripts. We assigned reads to this custom transcriptome using Salmon v.1.3.0[75] with parameters "-l A --validateMappings". We obtained > 40 million aligned reads per sample. Next, we summarised the transcript counts to gene counts and used this as input for downstream analysis. Normalization and differential analysis were performed using the DESeq2 R package v.1.36.0[76]. Threshold for significance was set at an FDR-adjusted *p*-value < 0.05 and an absolute log2 fold change (FC) > 1. All genes that did not reach significance or did not pass the log2 FC cutoff were classified as stable. As input for the heatmap in Fig. 5d, we produced a matrix of regularised log-transformed gene counts applying the "rlog" function from DESeq2. Gene Ontology enrichment analysis was performed using the goseq R package v.1.48.0[77] and the resulting *p*-values were adjusted for multiple testing using the FDR method. KEGG enrichment analysis was produced using the ClusterProfiler R package v.4.4.1[78]. Gene Set Enrichment Analysis (GSEA) was conducted using the fgsea function from the fgsea R package v.1.22.0 with default parameters, except in the case of "Beta cell disallowed genes" in which we specified the expected direction of enrichment with the argument "scoreType = 'neg'". In all the above analyses, terms were considered statistically significant when adjusted *p*-values < 0.05.

**Electron microscopy**. Cell spheroids were collected, washed with PBS and fixed with 4% paraformaldehyde/ 0.5% glutaraldehyde (Sigma) mixture in 0.1 M phosphate buffer (PB) pH7.4 for 30 min at 4 °C and gentle agitation. Cells were then transferred to fresh fixation solution and maintained at 4 °C until secondarily fixed with 1% uranyl acetate and 1% osmium tetroxide. Cells were then dehydrated, embedded in Spurr's Resin and sectioned using Leica ultramicrotome (Leica Microsystems). Conventional transmission electron microscopy (TEM) images were acquired from thin sections using a JEOL-1010 electron microscopy equipped with an SC1000 ORIUS-CCD digital camara (Gatan).

**Mouse studies**. Adult (8–20 week old) normoglycemic male NOD *scid* gamma (NSG™) mice (catalog no 005557, Jackson Laboratories) were used as transplantation recipients.

Approximately 300 cell spheroids (reprogrammed cells or parental fibroblasts) or 150–200 human islets were transplanted into the anterior chamber of the eye (ACE)[58]. In brief, an incision was made in the cornea near the corneoscleral junction, and a cannula (0.4 mm internal diameter) loaded with spheroids/islets, connected to a 500 µL syringe, was introduced into the incision. Cells were carefully injected without damaging the iris. The omentum, subcutatenous space and kidney were used for transplantation of higher number of cell spheroids (between 3500 and 5000). For transplantation in the omentum and subcutaneous space, cell spheroids were preloaded in collagen/Matrigel scaffolds. Briefly, the cell-laden collagen/Matrigel hydrogel was prepared by first mixing 110 µl of a 4 mg/ml rat tail type I collagen (Corning, NY, USA) solution with 40 µl of Matrigel (Corning) and then adding the mix to 5TF cell spheroids (pelleted by gentle centrifugation) and poured onto a cylindrical 8 mm diameter × 1 mm thick PDMS mold (Dow Corning Sylgard 184 Silicone Elastomer). The hydrogel was polymerized for 20 min at 37 °C,

detached from the mold and maintained in tissue culture dishes with warm RPMI-1640 medium until transplant (usually 2–3 h). Constructs were placed on the omentum close to the duodenal-stomach junction. Alternatively, constructs were introduced in the abdominal subcutaneous space through a small (5 mm) incision. Transplantation in the kidney was performed following standard procedures[79]. NSG mice transplanted with fibroblast spheroids an/or non-transplanted NSG mice were used as controls in transplantation experiments.

To assess vascularization and cell viability in ACE implants, 5TF cell spheroids were labeled with the long-term tracer for viable cells Vybrant CFDA SE (Invitrogen) before transplantation. At day 10 post-transplantation, mice received an intravenous injection of RITC-dextran and in vivo imaging was used to assess functional vascularization and cell viability[58]. For the determination of glucose-induced insulin secretion, mice were fasted for 5–6 h and then injected intraperitoneally with glucose (3 g/Kg). Tail blood was collected before and after (20 min) the glucose challenge. Aqueous humor from mice with ACE implants was obtained at time of sacrifice and kept frozen until human insulin determination. Human insulin in plasma and in aqueous humor was determined using an ultrasensitive Human Insulin ELISA (Chrystal Chem).

The Animal Research Committee of the University of Barcelona approved all animal procedures. European and local guidelines (Generalitat de Catalunya) on accommodation and care of laboratory animal were followed.

**Immunofluorescence and morphometric measurements**. Cells grown in 2D were fixed with 4% (v/v) paraformaldehyde (PFA) during 15 min and incubated with blocking solution (0.25% (v/v) Triton, 6% (v/v) donkey serum, 5% (w/v) BSA in PBS for 1 h at room temperature. Slides were then incubated with primary antibodies diluted in PBS-triton 0.1% (v/v) containing 1% donkey serum overnight at 4 °C. 5TFcells grown in spheroids were fixed with 4% (v/v) paraformaldehyde (PFA) for 15 min at 4 °C, permeabilized with 0.5% (v/v) Triton in PBS for 20 min and blocked with 0.5% (v/v) Triton/ FBS 10% (v/v) in PBS during 1 h at room temperature. Slides were then incubated with primary antibodies diluted in blocking solution overnight at 4 °C. Eyes and omentum implants were fixed overnight in 2 and 4% (v/v) PFA, respectively, dehydrated with ethanol gradient, cleared with xylene and paraffin-embedded. 3 μm thick eye sections were used for standard immunofluorescence staining protocol.

Primary antibodies used were: Insulin (DAKO, 1:400, Fig. 2b); Insulin/C-PEP (Hybridoma Bank; 1:40, used in all figures); HLA (Abcam, 1:100), PCSK1 (Gene Tex; 1:100); KCNJ11, PTPRN, and NCAM1 (Santa Cruz, 1:50). The antigen-primary antibody immune complex was visualized with secondary antibodies conjugated to Alexa Fluor 488 (Jackson Immunoresearch, 1:250), Alexa fluor 555 (Molecular Probes, 1:400) or Alexa fluor 647 (Jackson Immunoreserach, 1:250). Cell nuclei were counterstained with Hoechst 33258 (SIGMA, 1:500). Fluorescent images were captured using a Leica TCS SPE confocal microscope.

For morphometric analysis, total ACE grafts were sectioned at 3 μm and distributed as serial sections onto two sets of 10 slides each. At least 10 sections per graft, 90μm apart, were used to quantify the number of HLA+/INS+ and transcription factor+/INS+ cells using ImageJ/Fiji (National Institutes of Health, Bethesda, MD, USA; http://rsb.info.nih.gov/ij/) software. In 10-day grafts, to determine the percentage of INS+/HLA+ cells, 600–2000 HLA+ cells per graft were counted; and to determine the percentage of transcription factor+/INS+ cells, 200–1000 INS+ cells per graft were counted.

**Statistics and reproducibiltiy**. Data are presented as mean ± standard error of the mean (SEM) from at least three independent reprogramming experiments, with one to four biological replicates per experiment. Significant differences between the means were analyzed by the two-tailed unpaired Student's $t$-test, one sample $t$-test or one-way ANOVA followed by Tukey's or Dunnett's multiple comparison tests as indicated in the figure legends. Statistical analysis was performed with GraphPad Prism 8.00 and Microsoft Office Excel 2007 and differences were considered significant at $P < 0.05$. No methods were used to determine whether the data met assumptions of the statistical approach (e.g., test for normal distribution).

**Reporting summary**. Further information on research design is available in the Nature Portfolio Reporting Summary linked to this article.

## Data availability

RNA sequencing data are deposited in the Gene Expression Omnibus database under accession code GSE210075. Source data for main and supplementary figures are provided in Supplementary Data 2. All other data are available from the corresponding author upon reasonable request.

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

## Acknowledgements

We are indebted to Lidia Sanchez, Yaiza Esteban (IDIBAPS, CIBERDEM), Adriana Viladegut and Nawelle Dogheche (IDIBAPS) for their help in specific experiments, to Anna Soler (Hospital Clinic) for karyotyping the reprogrammed cells, and to Chris Newgard and Hans Hohmeier (Duke University) for critical reading of the manuscript. We thank the Functional Genomics Platform at IDIBAPS, and the Advanced Optical Microscopy and the Electronic Microscopy Units of the Technological Centers of the University of Barcelona (CCiTUB) for their help in sample processing and analyses. The monoclonal antibody against C-peptide was obtained from the Developmental Studies Hybridoma Bank developed under the auspices of the NICHD and maintained by The University of Iowa. This work has been supported by the following grants: PI19/00896 (to RGa, RGo) integrated in the Plan Estatal de I + D + I and cofinanced by ISCIII-Subdirección General de Evaluación and Fondo Europeo de Desarrollo Regional (FEDER-"A way to build Europe"); 120230 (to R.Ga) and 121430/31/32 (to N.M.) from La Fundació La Marató de TV3; and EFSD/JDRF/Lilly Type 1 Program 2017 (to R.Ga) from the European Foundation for the Study of Diabetes. The research leading to these results has received funding from Fundación DiabetesCERO (RGa) and from CIBER-Consorcio Centro de Investigación Biomédica en Red- (CB07/08/0009), Instituto de Salud Carlos III, Ministerio de Ciencia e Innovación.

## Author contributions

Conceived and designed the experiments: M.F. and R.Ga. Performed the experiments: M.F., A.G., E.P., N.T., H.F., R.F.R., S.C., and M.R.R. Provided human islets: C.B. and A.W. Provided materials: L.C., J.R., N.M., and A.N. Analyzed and discussed the data: J.M.S., L.P., M.F., C.E., J.V., R.Go, R.Ga. Wrote the manuscript: M.F. and R.Ga.

## Competing interests

The authors declare no competing interests.
