## [Peer Review File · Communications Biology]

Reviewers' comments:

Reviewer #1 (Remarks to the Author):

This submitted manuscript by Gasa and colleagues describes an approach to direct reprogram human dermal fibroblast cells into beta cells using 5 pancreatic transcription factors. This study described and enlightened the utility of transcriptional reprogramming of adult somatic cells without pluripotency induction into trans lineage cells such as pancreatic beta cells. Although the presented approach in this study is quite approachable if autologous cell transplantation is concerned without the need for immunosuppression and pluripotency risk and off-target issues, the use of adenoviral reprogramming in somatic cells and their acceptance in clinical application is still debatable. The study henceforth shows potential limitations from a clinical perspective. A subsection in the discussion for this caveat should be discussed.

Additionally, the study design also shows several more major concerns and limitations to access the data quality. My comments are as follows:

Study design and methods:

1. Although authors describe and test reprogramming fibroblast cells from autoimmune T1D patients, this could significantly influence or restrain the potential efficacy of transcriptional reprogramming and beta-cell functional maturation due to inherited genetic and genomic errors. As such, authors should test and demonstrate trans-differentiation efficacy using non-autoimmune patient-derived cells, in comparison. This is critical to assess the contribution of abnormal genetic load and pancreatic genomic dysregulation carryover from type-1 diabetes.
2. Authors must include the clinical history of patients.
3. How did the authors confirm fibroblast cell isolation and culture? What kind of cells do they isolate and expand? Flow characterization must be provided.
4. Is it not clear what authors did to transfect the cells post-Ad-PNM infection, whether they infected cells with both Pax4 and Nkx2.2 viruses separately one after the other or in combination? More clarity in methods is required.
5. Were the cells expanded post-infection? If yes, then how long? If not, then how authors were able to perform all the experiments as described with only 1.5 million cells? Considering that the repeated viral transduction would be highly cytotoxic and stressful, they might have been left with only minimal cells to work with. I am unable to understand the capacity to perform experiments without scalability. Authors must revise methods in more detail.
6. Authors describe that they generated spheroids from transduced cells after the last reprogramming and solely used them for the purpose of transplantation, while they perform all in-vitro diagnostic tests (Q-PCR and IHC) from cells cultured in plates without clustering. This could be a potential caveat since this is well known in the field that 3D clustering will induce and significantly enhance the expression of key pancreatic transcription factors on its own. Why would the author have tested cells without clustering and transplanted cells after clustering? Do they expect to see a difference in gene and protein expression? Will clustering enhance the cell maturity and efficacy of islet-like cluster generation? These questions are currently unanswered from performed methodology.
7. In transplantation procedures, the authors describe that they implanted cells at three sites in a single mouse: eye, kidney, and SubQ/ omental sheet. Why? Why did the authors not consider one transplantation site? Further, why do authors assess cells from the eye chamber only and why not from kidney and SubQ spaces? It is important to have data from the other two sites as well to check the quality and maturation of transplanted cells. This raises major concerns.
8. Authors inconsistently reported the dosing of iB-like cells (1200-1800 cells/clusters) for 300 clusters transplanted in the eye, (750-1000 cells/ cluster) transplanted elsewhere and (1000 cells/ cluster in legends). I am confused which one is correct? Why would they report different cell# / clusters if the cells are coming from the same prep and clustering experiment? Also, as per calculation 300 clusters with 1800 cells/cluster is only 540,000 cells which is not enough detectable insulin or c-peptide in the mouse.

9. Authors transplanted only 200 human islets per eye, I am not convinced if the cell mass is enough to measure human c-peptide, as what authors have reported. Mice are resistant to human insulin especially if they are normoglycemic when transplanted.
10. It is not clear if the authors have transplanted diabetic or non-diabetic mice. Authors must report the mice status and measure mice c-peptide in parallel.
11. The authors reported that they performed IPGTT on mice on day 10 PTx, however, they did not show any data for this.
12. N=3 mice for transplantation studies are not sufficient to demonstrate the effect. N's must be increased.

Results:

13. In figure1, Viral transduction efficiency and quantification are missing. Authors must supply the efficacy of transduction and potential cell loss to assess the cell yield.
14. Vector information must be supplied in supplementary data.
15. How many infected cells are expressing the TFs and for how long? Immunofluorescent and quantification for each TFs must be performed to confirm the quality of cells yielded.
16. In my opinion mRNA levels for each of these TFs in transduced cells are extremely lower compared to human islet/ pancreatic cells. This raises the serious concern over reprogramming efficiency and the effect being sustained for trans differentiation activation.
17. In fig-1 Human islet control is missing from each of the mRNA expression assays. The author must compare it with human islet expression and perhaps their cell expression will be negligible in comparison to donor islet expression.
18. Fig-1-f shows the data 1000-fold less INS expression in transduced cells than human islets. Such level of gene expression is insufficient to induce reprogramming and b-like cell trans-differentiation?
19. Differentiation control from i-B-like/ human islets must be quantified using flow cytometry to demonstrate the extent of cell maturation and differentiation using an Ad virus cocktail.
20. Transgene expression for PNM is impressive due to viral induction, while the same for Pax4 and NKX 2.2 and others are extremely low. As such authors are considering fibroblast cells as relative control, this will still show induction. I urge you to compare this with human pancreatic cells.
21. In fig-2 gain, human islet control is missing for Ins Gcg, Nkx6.1 and pax 6 expression.
22. Although the IF images support c-peptide induction in cells post 5TF transduction, the significant decline in the viability of cells is a major concern and essential to raise questions to obtain enough cell numbers to perform cell assays and transplantation studies.
23. Fig-3 a human islet control is missing.
24. Although 5Tfs are induced, not much profound effect (fold change) in gene expression is observed compared to PNM alone.
25. Why Pax4 expression is lower in 5TFs compared to PNM – Fig-3a?
26. In Fig-4a, 35% of cells show response to glucose and KCl while the majority 30+35% are non-responsive, suggesting only a fraction of cells are only showing mild function. Again, donor islet controls are missing.
27. In fig-5a protocol, cells moved into 3D for 3 days, authors should mention culture media and composition for this clustering induction in methods.
28. IHC for clusters should be performed more reliably with multiple other makers. Does the author propose that they are making only b-like cells, while they show gene expression for Gcg Sst and other makers?
29. Q-PCR data for the transition of cells from 2D to 3D is ineffective and shows a mild difference in TF induction and ins expression, I'm not convinced if the culture condition is appropriate for 3D induction.
30. Transplantation studies in mice are not well-planned. Proper graft IHC characterization for all islet hormones should be performed.
31. Figure 6D and F shows an in-vivo comparison with fibroblast cells. Did authors transplant non-transduced cells in mice? If yes, this is missing from methods.
32. HI data (Fig-6f) clearly says half of the mice show no to mild release while half show nice insulin release in aqueous humour. Why does the author show this kind of discrete separation? Does this imply the quality and inconstancy of transduced cells? I am doubting the statistical calculations the

tight CVs are shown while having such a huge separation.

33. Mild changes in plasma HI data over 15- 30 days transplantation, showing limited cell maturation over time. Considering that authors implanted 4000 clusters i.e nearly 5 million beta cells, one should expect much more signals and maturation at 30 days PTx. Perhaps they need long-term follow-up for 20 weeks-25 weeks, 4 weeks is not enough.

Minor comments:

1. It is inevitable to perform cell quality assessment using flow cytometry.
2. Methods are not written in detail. More information is required
3. Inconstancies in reporting cell number, clusters and no of iB-like cells. Should be carefully verified.
4. Mice studies should be more well planned with long-term follow-up.
5. Empty Ad vector controls should be used as the negative control in the entire study.

Reviewer #2 (Remarks to the Author):

The study by Gasa and colleagues presents a new method to induce insulin-secreting cells from human fibroblasts. In vitro and in vivo analyses suggest a degree of molecular and functional resemblance of the cells with pancreatic beta cells although the expression levels of beta genes and the rate of insulin secretion are well below that of beta cells. Nevertheless, the method is an improvement from prior studies to convert fibroblasts. The multiple analytical tools applied, including calcium imaging and transplantation, yielded insight on the properties of these cells. Some concerns and suggestions below.

1. Fig. 2. What was the rationale for expressing Pax4 followed by NKX2.2? any supporting data? how was the optimal viral MOI determined? what was the percentage of cherry+ cells that were cPPT+?

2. uIU/ml or pmol/L are commonly used to show insulin levels. For ease of comparison with other publications, please consider using these units.

3. Fig. 5D. Is this the total insulin amount collected over how long a period? How do they compare with human islets?

4. Fig. 6G, it would be informative to have images of grafts in kidney, subcute and omentum. Were the grafts tracked beyond 30 days? Did insulin levels change over time?

5. Fig. S4B, it seems many cPPT+ cells did not express Pdx1 or Mafa. Please quantify. Any explanations here?

6. To have a better understanding of molecular similarities and differences between fibroblast-derived beta-like cells with pancreatic beta cells, gene profiling is much preferred over qPCR of a small number of genes.

7. Foreskin and adult fibroblasts were mentioned in the Method but it is not clearly what data were generated from what cells.

POINT-BY-POINT RESPONSE TO REVIEWERS

Manuscript #: COMMSBIO-21-3291-T

Reviewer #1 (Remarks to the Author)

This submitted manuscript by Gasca and colleagues describes an approach to direct reprogram human dermal fibroblast cells into beta cells using 5 pancreatic transcription factors. This study described and enlightened the utility of transcriptional reprogramming of adult somatic cells without pluripotency induction into trans lineage cells such as pancreatic beta cells. Although the presented approach in this study is quite approachable if autologous cell transplantation is concerned without the need for immunosuppression and pluripotency risk and off-target issues, the use of adenoviral reprogramming in somatic cells and their acceptance in clinical application is still debatable. The study henceforth shows potential limitations from a clinical perspective. A subsection in the discussion for this caveat should be discussed.

Additionally, the study design also shows several more major concerns and limitations to access the data quality. My comments are as follows:

Study design and methods:

1. Although authors describe and test reprogramming fibroblast cells from autoimmune T1D patients, this could significantly influence or restrain the potential efficacy of transcriptional reprogramming and beta-cell functional maturation due to inherited genetic and genomic errors. As such, authors should test and demonstrate trans-differentiation efficacy using non-autoimmune patient-derived cells, in comparison. This is critical to assess the contribution of abnormal genetic load and pancreatic genomic dysregulation carryover from type-1 diabetes.

Response: All the experiments in the submitted manuscript were performed with control human foreskin fibroblasts. We acknowledge that in the Methodology Section of the submitted transcript we mistakenly also mentioned fibroblasts of T1D patients. This has been corrected in the revised manuscript.

2. Authors must include the clinical history of patients.

Response: No patients are used in this study.

3. How did the authors confirm fibroblast cell isolation and culture? What kind of cells do they isolate and expand? Flow characterization must be provided.

Response: HFF1 fibroblasts (used in all main figures and in Supplemental figures were prepared by Núria Montserrat (co-author in the manuscript) following standardized protocols routinely used at the Stem Cell Bank of the Center of Regenerative Medicine (<https://p-cmrc.cat/research>) to generate iPSC. On the other hand, HFF2 fibroblasts, which were used to validate the reprogramming protocol established with the HFF1 line, were purchased from ATCC (Supplementary Figure S5).

4. Is it not clear what authors did to transfect the cells post-Ad-NPM infection, whether they

infected cells with both Pax4 and Nkx2.2 viruses separately one after the other or in combination? More clarity in methods is required.

Response: As depicted in the diagram of the reprogramming protocol provided in Figure 3A (Figure 2B in original submission), Pax4 and Nkx2-2 were added sequentially. In order to make this point clearer, we have re-written the description of the reprogramming technique in the Methods section of the revised manuscript.

5. Were the cells expanded post-infection? If yes, then how long? If not, then how authors were able to perform all the experiments as described with only 1.5 million cells? Considering that the repeated viral transduction would be highly cytotoxic and stressful, they might have been left with only minimal cells to work with. I am unable to understand the capacity to perform experiments without scalability. Authors must revise methods in more detail.

Response: We grow parental fibroblasts prior to reprogramming, but we do not expand reprogramming-in-progress cells. We initially plate 1.5 million fibroblasts per 10 cm plate (amount of cells varies according to the area of growth surface of the plate/flask). Reprogramming begins 16-24h after plating, or when cells have reached 80% confluence. NPM causes a blockage of cell growth but not a reduction in viability (see Figure 3C,D). By the end of the protocol, from the 1.5 million fibroblasts that we initially plate, we generate 1200-1500 clusters of 1000 cells/cluster. Depending on how many insulin-producing cells are needed to complete an experiment, we determine the number of parental fibroblasts to plate for that experiment.

6. Authors describe that they generated spheroids from transduced cells after the last reprogramming and solely used them for the purpose of transplantation, while they perform all in-vitro diagnostic tests (Q-PCR and IHC) from cells cultured in plates without clustering. This could be a potential caveat since this is well known in the field that 3D clustering will induce and significantly enhance the expression of key pancreatic transcription factors on its own. Why would the author have tested cells without clustering and transplanted cells after clustering? Do they expect to see a difference in gene and protein expression? Will clustering enhance the cell maturity and efficacy of islet-like cluster generation? These questions are currently unanswered from performed methodology.

Response: The effects of 3D clustering were evaluated before any transplantation experiment was carried out. This characterization included: gene expression analysis by qPCR of selected genes, immunostaining for C-PEPTIDE, glucose-induced insulin secretion and electron microscopy. As the reviewer indicates, 3D culturing improved differentiation markers as compared to 2D cultures. All these information was provided in Figure 5 of our first submission. In the revised manuscript, these data are provided in Figures 5 and 6. Please take note that the revised manuscript contains brand-new gene-profiling data (Figure 5D-G).

7. In transplantation procedures, the authors describe that they implanted cells at three sites in a single mouse: eye, kidney, and SubQ/ omental sheet. Why? Why did the authors not consider one transplantation site? Further, why do authors assess cells from the eye chamber only and why not from kidney and SubQ spaces? It is important to have data from the other two sites as well to check the quality and maturation of transplanted cells. This raises major concerns.

Response: Spheroids were implanted in a single location in each animal. We first performed transplants in the anterior chamber of the eye because of the benefits of this site for islet

engraftment including (a) rapid revascularization (dense vascular network in the iris), (b) easy supply of sympathetic and parasympathetic fibers (dense eye innervation) and (c) compatibility with longitudinal in vivo imaging (recently reviewed in Llegems E, Berggren P:O, *Frontiers in endocrinology* 2021). However, our studies in the ACE showed that more cells were required if we wanted to detect human insulin in plasma. Thus, we moved to other transplantation sites able to accommodate 3500–5000 clusters (ACE can only accommodate 300–400 clusters). These sites were: subcutaneous, omentum and kidney. The reviewer's comment has made us aware that using many sites may confuse readers and take them away from the research' main objective, which was to determine whether grafts released insulin in response to glucose in vivo. For this reason, we have carried out more transplants in the omentum, which was the site that showed us the most promise, in order to prepare the revised paper. Figure S9 and Table S2 of the revised manuscript show these results. Note that results from subcutaneous and kidney sites are still shown in Table S2 of the revised manuscript. We also present new representative images of reprogrammed INS+ cells identified in 30-day omentum grafts in Figure S9.

8. Authors inconsistently reported the dosing of iB-like cells (1200-1800 cells/clusters) for 300 clusters transplanted in the eye, (750-1000 cells/ cluster) transplanted elsewhere and (1000 cells/ cluster in legends). I am confused which one is correct?

Response: We appreciate the reviewer bringing up the discrepancies between the cell numbers stated in our manuscript's Methods and Figure Legends. We have corrected them in the revised manuscript. In brief, spheroids of 1200-1800 cells generated in 96-well plates were used for gene expression, immunofluorescence and in vitro insulin secretion assays. On the other hand, for transplants, which required a higher number of clusters, these were generated in Aggrewell-400 plates. In order to enable optimal spheroid formation in the microwells of these plates, which are 400 μ M in diameter, we reduced their size to 1000 cells in this case.

Why would they report different cell# / clusters if the cells are coming from the same prep and clustering experiment?

Response: It is not possible for all the cells utilized in the experiments of this manuscript to have come from the same clustering experiment.

Also, as per calculation 300 clusters with 1800 cells/cluster is only 540,000 cells which is not enough detectable insulin or c-peptide in the mouse.

Response: The reviewer is correct. In fact, insulin levels shown in Figure 7F of the revised manuscript (Figure 6F in original submission) correspond to levels detected in the aqueous humor (fluid that fills the ACE, where clusters are transplanted, approx. 5 μ l recovered per eye), not in blood. In order to detect plasma insulin levels we had to raise the number of transplanted clusters to between 3500 and 5000 (Figure S9 of revised manuscript).

9. Authors transplanted only 200 human islets per eye, I am not convinced if the cell mass is enough to measure human c-peptide, as what authors have reported. Mice are resistant to human insulin especially if they are normoglycemic when transplanted.

Response: We are not sure what the reviewer is referring to with the statement that "mice are resistant to human insulin". We report human insulin levels (we did not measure C-PEP) in the ocular aqueous humour of eyes transplanted with human islets in Figure 7F of the revised manuscript (Figure 6F in original submission). From these values, we calculate that

300 5TF cell clusters produce on average approximately 20-fold less insulin than 150-200 human islets at day 10 post-transplant.

10. It is not clear if the authors have transplanted diabetic or non-diabetic mice. Authors must report the mice status and measure mice c-peptide in parallel.

Response: We only used normoglycemic NSG mice (stated in the Methods and Results section of the revised manuscript).

11. The authors reported that they performed IPGTT on mice on day 10 PTx, however, they did not show any data for this.

Response: We did not perform an IPGTT. We performed *in vivo* GSIS (testing plasma insulin levels before and 20 minutes after a glucose bolus). This is shown in Figure S9 of the revised manuscript (Figure 6G in the first submission).

12. N=3 mice for transplantation studies are not sufficient to demonstrate the effect. N's must be increased.

Response: To avoid confusion, we have now raised the number of transplants in the omentum (from n=3 to n=10). These data is presented in Figure S9 and Table S2. Results from other transplant sites are also provided in Table S2, as detailed in response to point 7.

Results:

13. In figure1, Viral transduction efficiency and quantification are missing. Authors must supply the efficacy of transduction and potential cell loss to assess the cell yield.

Response: The estimated efficiency of viral transduction was >80% (as determined by Cherry immunofluorescence). It is reported in the text and visible in images in Figure 1A. Good viral infection is also shown in Figure S1 where we portray the improvement in viral transduction when adding a transfection reagent to the adenovirus-containing mix.

14. Vector information must be supplied in supplementary data.

Response: Adenoviral vectors used in the study had been previously described. We provide bibliographic references for all of them. The adenovirus encoding Mafa was purchases from a commercial source as indicated in the Methods Section.

15. How many infected cells are expressing the TFs and for how long? Immunofluorescent and quantification for each TFs must be performed to confirm the quality of cells yielded.

Response: In agreement with Cherry fluorescence shown in Figure 1A, we detected expression of the adenovirus-encoded TFs in 80% of the cells. We provide the figure below, which represents 3-day clusters, as an illustration. Please note the presence of Pdx1 (encoded by Ad.NPM) and Nkx2.2 (encoded by Ad.Nkx2.2) in most cells. Most importantly, both factors are found co-expressed. Transgene expression declines over time but it is still clearly detectable after 21-days in culture (Figure 3G) and 10 days post-transplantation (Fig.S7).

16. In my opinion mRNA levels for each of these TFs in transduced cells are extremely lower compared to human islet/ pancreatic cells. This raises the serious concern over reprogramming efficiency and the effect being sustained for trans differentiation activation.

Response: It is true that native (endogenous) mRNA levels for the reprogramming TFs are lower than in human islets, ranging from 4-9-fold lower for *NGN3* and *PAX4* (note that these TFs are marginally expressed in adult human islets), to 70-fold for *PDX1* and 300-fold for *NKX2-2* (Figure 5C). We must keep in mind; nevertheless, that the concurrent expression of TFs produced by adenoviruses may impact regulation of the endogenous gene. In fact, transcriptional auto-regulation is frequently observed throughout development. It was unexpected that, provided that adenoviral vectors do not integrate in the host genome, adenoviral-encoded genes persisted for a long period of time (see point 15). Intriguingly, similar findings were reported in trans-differentiation protocols from human ductal to insulin-secreting cells that also used adenoviruses as delivery vectors (Lee et al. eLife 2013; 2:e00940). The persistence of adenoviral expression may be explained by the fact that these cells stop growing after NPM expression. We realize that this is a limitation of the present experimental model and that it might necessitate the use of alternative transgene delivery methods. However, these are beyond the scope of the present paper. We have addressed the possible impact of persistent transgene expression in the of the revised manuscript.

17. In fig-1 Human islet control is missing from each of the mRNA expression assays. The author must compare it with human islet expression and perhaps their cell expression will be negligible in comparison to donor islet expression.

Response: In Figure 5C of the revised manuscript, we present a comparison of the mRNA levels of 19 genes in 3D 5TF cell spheroids and human islets, expressed as fold-change. We acknowledge that the expression of these genes is lower in reprogrammed cell clusters relative to human islets, ranging from 2-fold for *NCAM* to 450-fold lower for *ABCC8*. These results indicate that our present 5TF-3D reprogramming protocol needs to be further optimized. We are currently investigating this in the lab, and preliminary findings show that it

is feasible with minor adjustments like lengthening the culture period or adding soluble molecules. We intend to present these and other data in a subsequent manuscript.

18. Fig-1-f shows the data 1000-fold less *INS* expression in transduced cells than human islets. Such level of gene expression is insufficient to induce reprogramming and b-like cell trans-differentiation?

Response: We understand that the reviewer refers to *INS* expression of cells harvested seven days following the addition of the NPM factors (Figure 1D in the revised paper and in original submission). After completion of the 5TF-3D protocol, *INS* levels were 6-fold higher than with tNPM alone, reaching levels that were 170-fold those of human islets (Figure 5C of the revised manuscript). We consider this result a substantial accomplishment and, to the best of our knowledge, it is the first time to be achieved using human fibroblasts as cell source. While we understand that there is room for improvement and are very interested in doing so, we feel this is outside the scope of the current manuscript, which aims to demonstrate the viability of converting human fibroblasts toward the beta cell lineage via TF-based direct reprogramming. As of right now, we cannot and do not claim that these cells are equivalent to primary beta cells.

19. Differentiation control from i-B-like/ human islets must be quantified using flow cytometry to demonstrate the extent of cell maturation and differentiation using an Ad virus cocktail.

Response: We have determined the percentage of *INS*⁺ cells by immunofluorescence staining. We found 67.9±6.2% of cells in the 2D-5TF cultures were *INS*⁺ at day 10 (reported in the Results section). We now also provide the percentage of *INS*⁺ cells in the grafts ten days after transplantation: 43.5±2.8% *HLA*⁺*INS*⁺/*HLA*⁺ cells (Figure 7E). We also provide a new figure showing the absence of glucagon or somatostatin staining in 5TF cell clusters (Figure S4).

20. Transgene expression for NPM is impressive due to viral induction, while the same for Pax4 and NKX 2.2 and others are extremely low. As such authors are considering fibroblast cells as relative control, this will still show induction. I urge you to compare this with human pancreatic cells.

Response: We thank the reviewer for bringing up this point. After reading the reviewer's comment, we realized that we had not included expression levels for the *Nkx2.2* and *Pax4* transgenes in our first submission. We believe that the reviewer may have referred to endogenous expression of these two genes in NPM-transduced cells (Figure 1E in original paper, Figure 1F in revised manuscript). We now provide transgene expression levels for *Nkx2.2* and *Pax4* in Figure 2A of the revised manuscript.

Accurate comparison of gene expression levels with human islets can only be made for *NKX2-2* as the transgene is also human and qPCR components and conditions are the same. As shown in the figure below, *NKX2-2* transcript levels were seven times higher in reprogrammed cells at day three following adenoviral infection than in human islets. Transgene expression gradually declines over time, therefore it would be anticipated to approach levels similar to those of human islets. One factor we need to take into account is whether the TF levels in differentiated cells are different from those required to encourage differentiation throughout development.

21. In fig-2 gain, human islet control is missing for *Ins Gcg*, *Nkx6.1* and *pax 6* expression.

Response: Human islet gene expression data is shown in Figure 1D (INS) and Figure 5C (INS and other beta cell genes), where it is presented as fold-change difference between expression of the indicated genes in human islets relative to 5TF cell spheroids at day 11 of the protocol. Since it would be redundant, we do not provide human islet gene expression data at prior stages when the reprogramming technique was being developed.

22. Although the IF images support c-peptide induction in cells post 5TF transduction, the significant decline in the viability of cells is a major concern and essential to raise questions to obtain enough cell numbers to perform cell assays and transplantation studies.

Response: After NPM introduction, we observed a blockade in cell proliferation as early as day 4 post-NPM and no changes in viability as measured by MTT assay either at day 3 or day 7 of the protocol.

We did observe a decline in the MTT assay at day 10 (reported in Figure 2E of the original submission), which we understood to be a result of the fewer cells in the wells since they stopped proliferating (note that we measured MTT reduction per well, not per number of cells). In support of this interpretation, we do not observe significant upregulation of stress and apoptotic genes as determined by qPCR or caspase 8 activation assay (extrinsic apoptotic pathway) in 5TF cells (graphs on the right;

mean \pm SE, n=3-5; TGH refers to 1 μ M thapsigargin used as ER stressor). Additionally, we did not identify gene set enrichment of cell survival/death or related processes in our RNA-seq data.

Therefore, in the revised paper, we provide the results of the MTT assays at days 4 and 7 as readout of cell viability (new Fig. 3C, D). We have withdrawn MTT data on day 10 as it was misleading. We will be happy to provide the above gene expression data as a supplementary figure upon the reviewer's request.

23. Fig-3 a human islet control is missing.

Response: see response to point 21.

24. Although 5Tfs are induced, not much profound effect (fold change) in gene expression is observed compared to NPM alone.

Response: The reviewer is correct that expression of some genes does not change much between NPM and 5TF (e.g. *PCSK1*, *PTPRN*, *MAFB*, *NCAM1*). However, some other genes show a considerable increase in 5TF relative to NPM (e.g. *INSM1*, *GLP1R*, *HNF1B*), sometimes from undetectable levels (*NKX6-1*, *ABCC8*, *PAX6*). Importantly, the expression of other genes, including *GCG* and *PCSK2*, both of which are strongly expressed in alpha cells, is suppressed in 5TF relative to NPM. These results lead us to suggest that the 5TF procedure reprograms cells more closely to the beta cell lineage than NPM alone.

25. Why Pax4 expression is lower in 5TFs compared to NPM – Fig-3a?

Response: This is an interesting observation. Pax4 is a direct target of Neurog3 and is expressed in developing beta cells; however, it is absent from mature islets (Wang et al. *Dev Biol* 2003; 266: 178-189). To the best of our knowledge, the mechanisms regulating Pax4 suppression as beta cells mature are not described. Our observation that endogenous *PAX4* mRNA levels are lower in 5TF cells than in cells expressing only the NPM factors suggests that exogenous Nkx2.2 and/or Pax4 itself may inhibit endogenous *PAX4* expression. In support of the first possibility, we observed that ectopic Nkx2-2 downregulates NPM-induced *PAX4* expression in human fibroblasts (see graph, note that *PAX4* gene expression is induced by NPM relative to controls, C, but blocked when Nkx2.2 is added either 1,3 or 6 days after NPM).

26. In Fig-4a, 35% of cells show response to glucose and KCl while the majority 30+35% are non-responsive, suggesting only a fraction of cells are only showing mild function. Again, donor islet controls are missing.

Response: Figure 4A shows that 35% of the assayed cells responded to both glucose and KCl while 27% responded to depolarization but not glucose. Only 35% of the cells showed no changes in response to any of these stimuli. These findings are comparable to those for islet cells derived from stem cells, where the overall fraction of glucose-responsive cells ranges from 40 to 70% (Pagliuca et al. *Cell* 2014; 159: 428-439, Fig.2; Balboa et al. *Nat Biotech* 2022; 40:1042-1055).

The available literature shows that human beta cells are highly heterogenous in their calcium responses to glucose (Quesada I et al. *Diabetes* 55, 2463-2469, 2006; Kenty J. *PlosOne* 2015, 10: e0122044; Balboa et al. *Nat Biotech* 2022; 40:1042-1055). Similarly, we have found that mouse MIN6 cells exhibit a broad array of calcium responses to glucose (shown in graphs on the right, corresponding to 6 individual cells, not included in revised manuscript). This heterogeneity makes it challenging to draw any significant conclusion from simply comparing response patterns from a limited number of cells. Because of this, the limited availability of tissue and the high variability among human islet preparations, we pondered that a human islet control was not necessary in this particular experiment. We are

confident that our data as presented clearly shows that a percentage of reprogrammed cells exhibit calcium responses to glucose while parental fibroblasts don't. How similar this response is to that of human beta cells is indeed a very relevant question that requires a comprehensive assessment of calcium dynamics that we feel is beyond the scope of our present study.

27. In fig-5a protocol, cells moved into 3D for 3 days, authors should mention culture media and composition for this clustering induction in methods.

Response: Following the reviewer's request, in the Methods section of the revised manuscript, we offer a better explanation of the protocol.

28. IHC for clusters should be performed more reliably with multiple other makers. Does the author propose that they are making only b-like cells, while they show gene expression for Gcg Sst and other makers?

Response: Following the reviewer's request, we have performed immunostaining for GCG and SST in 5TF clusters. As shown in new Figure S4, we could not detect either hormone in 5TF cell clusters.

29. Q-PCR data for the transition of cells from 2D to 3D is ineffective and shows a mild difference in TF induction and ins expression, I'm not convinced if the culture condition is appropriate for 3D induction.

Response: We agree that some of the changes at the gene expression level are modest, although expression of key function genes such as *ABCC8* and *KCNJ11* genes are increased 4-5 fold by only 3 days in 3D as compared to 2D culture. Most significantly, switching to 3D culture was sufficient to enable a moderate but significant amount of glucose-induced insulin secretion. These results imply that human fibroblasts are amenable to direct transformation into insulin-producing cells that secrete insulin, even if we recognize that there is potential for improvement. These discoveries are significant. They go against earlier conclusions and can potentially open up new avenues for research into beta cell replacements.

30. Transplantation studies in mice are not well-planned. Proper graft IHC characterization for all islet hormones showed be performed.

Response: In response to the reviewer's comment we provide more information of the transplantation studies in mice in our revised manuscript. On one hand, we have performed additional omentum transplants (see response to point 7). On the other hand, we show images and report morphometric quantification of reprogrammed cells at day 10 and 30 post-transplantation. These data are provided in Figures 7, S6-S9 and Table S2 of the revised manuscript. Please note that we use *INS* gene expression and INS immunostaining in harvested grafts as readout of reprogramming. In the revised manuscript we also show negative immunostainings for glucagon and somatostatin in 5TF clusters before transplantation (Figure S4). Based on these results we found it unnecessary to repeat these stainings in the harvested grafts. See response to point 33 for further explanation regarding long-term follow up studies.

31. Figure 6D and F shows an in-vivo comparison with fibroblast cells. Did authors transplant non-transduced cells in mice? If yes, this is missing from methods.

Response: Yes, we used non-transplanted and parental fibroblasts as controls in our transplantation experiments in the ACE. We used non-transplanted mice as controls for

transplantation experiments in other sites. We have now mentioned these controls in the Methods section.

32. HI data (Fig-6f) clearly says half of the mice show no to mild release while half show nice insulin release in aqueous humour. Why does the author show this kind of discrete separation? Does this imply the quality and inconstancy of transduced cells? I am doubting the statistical calculations the tight CVs are shown while having such a huge separation.

Response: Like the reviewer, we were also puzzled by this observation. Unfortunately, we have not yet found a single convincing explanation for it. It may be attributed to variation in the amount of clusters correctly transported to the ACE, degree of revascularization, or technical difficulties with aqueous humor collection, a micro-technique, rather than reprogramming efficiency (measured by *INS* gene expression or INS+ immunostaining). These findings have been re-plotted using a more suitable graph type, and an additional human islet control has been included (new Fig.7F). Human insulin levels are significantly higher in eyes transplanted with 5FT spheroids relative to untransplanted eyes or eyes transplanted with control fibroblasts using a student's t-test. This is true when comparing all 5TF samples ($p=0,0002$) or when separating them between high expressers ($p=0,00005$) and low-expressers ($p=0,000003$).

33. Mild changes in plasma HI data over 15- 30 days transplantation, showing limited cell maturation over time. Considering that authors implanted 4000 clusters i.e nearly 5 million beta cells, one should expect much more signals and maturation at 30 days PTx. Perhaps they need long-term follow-up for 20 weeks-25 weeks, 4 weeks is not enough.

Response: We agree with the reviewer that long-term follow up of the transplants would be highly informative. However, after performing insulin immunostaining in 30-day grafts, we discovered a general decline in the number of INS+ cells (these images are provided in new Figure S7), preventing us from extending the follow-up period. At days 10 and 30, we have evaluated the percentage of HLA+/C-PEP+ in ACE grafts, and found that, although average values are only slightly lower at day 30, there is higher variability at day 30 than at day 10, going from almost 0 to >60% in 30-day grafts (graph on the right). These data are provided in Figure 7 and Figure S8 of the revised manuscript.

We are currently trying to determine what causes poor cell survival beyond 2 weeks post-transplantation. We speculate that cells die because of apoptosis. Preliminary TUNEL assays, however, reveal cell death at day 2 as expected, which rapidly declines at days 4 and 10 (AGA, unpublished observations). It should be noted that cell loss after 2-week transplantation was previously reported in human duct cell-derived beta cells (Lee et al. eLife213; 2:e00940). Like us, these authors found that adenoviral expression persisted in reprogrammed cells. Whether loss of reprogrammed cells is due to persistent adenoviral expression, reprogramming itself or implantation issues will need to be deciphered before we can conduct long-term transplantation experiments. We make reference to this problem in the discussion section of the revised manuscript.

Minor comments:

1. It is inevitable to perform cell quality assessment using flow cytometry.

Response: While flow cytometry is highly valuable we do not think that it would add more information as compared to quantification by immunostaining in the present manuscript.

2. Methods are not written in detail. More information is required 3. Inconstancies in reporting cell number, clusters and no of iB-like cells. Should be carefully verified.

Response: We have addressed the reviewer's concern and re-written the Methods section in more detail, and fixed the inconsistent cell numbers.

4. Mice studies should be more well planned with long-term follow-up.

Response: We agree with the reviewer that long-term follow-up would be informative. However, as detailed in prior point 33, enhanced survival of reprogrammed cells is first needed in order to address this question.

5. Empty Ad vector controls should be used as the negative control in the entire study.

Response: At the beginning of the study, we employed a control adenovirus expressing B-galactosidase and confirmed that it did not induce the *INS* gene in human fibroblasts (Figure 1D). In addition, as depicted in Figure S3, we have employed multiple viruses during the study and found that only the NPM factors delivered via the polycistronic vector induced significant *INS* gene activation. These observations support the specificity of our findings. Furthermore, given that we included five different proteins in our procedure, it was difficult to decide between a control vector that was empty or one or several that contained an unrelated protein. For all of these reasons, we decided to utilize fibroblasts as controls, which replicate the biological environment in which 5TF cells originally developed.

Reviewer #2 (Remarks to the Author):

The study by Gasa and colleagues presents a new method to induce insulin-secreting cells from human fibroblasts. In vitro and in vivo analyses suggest a degree of molecular and functional resemblance of the cells with pancreatic beta cells although the expression levels of beta genes and the rate of insulin secretion are well below that of beta cells. Nevertheless, the method is an improvement from prior studies to convert fibroblasts. The multiple analytical tools applied, including calcium imaging and transplantation, yielded insight on the properties of these cells. Some concerns and suggestions below.

1. Fig. 2. What was the rationale for expressing Pax4 followed by NKX2.2? any supporting data?

Response: We appreciate the reviewer's inquiry, which made us aware that we needed to clarify the rationale for using these additional TF in our reprogramming protocol. As a result, we have done so in the corresponding part of the revised paper. Both Pax4 and Nkx2.2 are expressed simultaneously in beta cell precursors where they cooperate to promote differentiation (Wang et al. Dev Biol 2003; 266: 178-189). The order was chosen mostly in accordance with earlier research (Gasa et al. PNAS 2004, 101: 13245-13250), which showed that Pax4 was one of the first genes to be induced by ectopic Neurog3, before Nkx2-2. Therefore, we decided that adding it first made sense. In addition, in preliminary experiments for the present study we had observed that Nkx2-2 reduced NPM-induced endogenous *PAX4* expression in

PNM

fibroblasts. These observations suggested that Nkx2.2 might be crucial to shut down Pax4 expression as beta cells differentiate. Given that Pax4 expression is transient throughout development and absent from mature beta cells, it made sense to add it third in our experiment.

how was the optimal viral MOI determined?

Response: In order to establish the ideal dose of the NPM adenovirus, we conducted early titration studies using *INS* gene expression (3-5 days after infection) as readout. An example is shown below. In this instance, we opted for the 25ul dose because boosting it twice did not significantly raise the levels of *INS* or the transgenes.

As for the Pax4 and Nkx2-2 adenoviruses, we performed similar titration experiments. In this case, the readout was *INS* gene expression at day 10 of the reprogramming protocol (*INS* was not induced in the absence of the NPM factors). Please take note that each time we utilized new adenovirus prep, we ran a titration experiment.

What was the percentage of cherry+ cells that were cPPT+?

Response: We did not quantify the percentage of Cherry+/*INS*+ cells.

However, we report that $67.9 \pm 6.2\%$ of cells in the 2D-5TF cultures were *INS*+ at day 10 (second Results section) and that $43.5 \pm 2.8\%$ of the HLA+ cells were *INS*+ in the grafts 10 days after transplantation (last Results section). Our typical infection efficiency is about 80% (see Figure 1A for Cherry, or image on the right showing

staining for Pdx1 –encoded by the NPM virus- and for Nkx2-2 –encoded by the Nkx2-2 virus- in 3 days 5TF cell clusters). Based on these numbers we estimate that 60% of the cells receiving the viruses turn on *INS* expression.

2. uIU/ml or pmol/L are commonly used to show insulin levels. For ease of comparison with other publications, please consider using these units.

Response: We have changed to pmol/L as suggested by the reviewer.

3. Fig. 5D. Is this the total insulin amount collected over how long a period? How do they compare with human islets?

Response: Figure 5D of the original submission (Figure 6C in the revised manuscript) depicts the amount of insulin secreted by 50 5TF clusters in 90 min. This is now indicated in the axis label. Similar numbers of cultured human islets secrete, on average, 40–70 times more insulin than the 5TF cluster, according to findings from our group (2mM: 128 ± 28 pmol/50 islets, n=6; 20mM: 398 ± 120 pmol/50 islets, n=6). These numbers, however, might not be exact as the data is derived from one human islet preparation and it is well known that donor and batch characteristics can greatly influence the performance of isolated human islets. We have tried to compare our values to those described by others, however it has proven difficult due to the broad array of assay and normalization methods used in the literature. Yet, we think that the range of 50-100-fold is likely accurate as human insulin levels in the aqueous humor of ACE transplanted with 300 5TF spheroids were approximately 20-fold (range between 20-140) those of 150-200 human islets (Figure 7F).

4. Fig. 6G, it would be informative to have images of grafts in kidney, subcutane and omentum. Were the grafts tracked beyond 30 days? Did insulin levels change over time?

Response: In response to the reviewers' request, we performed additional transplantation experiments in the ACE and omentum to perform immunostaining in grafts harvested at day 30. The number of INS+ cells that we found is drastically lower than for 10 day grafts irrespective of the transplantation site. Notably, the proportion of INS+/HLA+ cells is more uniform at day 10 than it was at day 30, where it is highly heterogeneous (see graph on the right, results from the ACE). We are unsure of the exact reasons for this variability. These data are provided in Fig.7 and Fig. S8 of the revised manuscript.

We are currently trying to determine what causes poor cell survival beyond 2 weeks post-transplantation. We speculate that cells die because of apoptosis. Preliminary TUNEL assays, however, reveal cell death at day 2 as expected, which rapidly declines at days 4 and 10 (AGA, unpublished observations). It should be noted that cell loss after 2-week transplantation was previously reported in human duct cell-derived beta cells (Lee et al. eLife213; 2:e00940). Like us, these authors found that adenoviral expression persisted in reprogrammed cells. Whether loss of reprogrammed cells is due to persistent adenoviral expression, reprogramming itself or implantation issues will need to be deciphered before we can conduct long-term transplantation experiments. We make reference to this problem in the discussion section of the revised manuscript.

Additionally, we would like to inform the reviewer we have expanded the section that shows the in vivo findings from our study. These data are provided in Figures 7, S6-S9 and Table S2 of the revised manuscript.

5. Fig. S4B, it seems many cPPT+ cells did not express Pdx1 or Mafa. Please quantify. Any explanations here?

Response: The reviewer is correct in pointing out that the co-expression of these TFs with INS in the photographs we included in our initial submission could hardly be noticed (Fig. S4). To aid in the viewing of staining, we have selected new photos of higher quality and provided separate images for each antibody (new Figure S7).

We have also performed morphometric quantifications, and we can now state that in 10-day ACE grafts, 20 to 30% of INS+ cells exhibit readily detectable positive staining for Mafa, Pdx1, or Nkx2-2 (table below and Figure S7).

Pdx1/INS+	29.8±2.6
Mafa+/INS+	20.0±1.8
Nkx2-2+/INS+	27.8±2.6

Although we were unable to differentiate between endogenous and exogenous protein, we speculate that strong TF staining seen in these INS+ cells might be due to sustained expression of the virally encoded reprogramming TF. We believe so because transgene transcripts are still detected in the harvested grafts (as example, we show *Pdx1* transcript level in Figure S7). Knowing that adenoviral products can persist after transplant raises the question of how long this might last and what impact it would have on survival of reprogrammed cells. Interestingly, similar findings were reported by others in reprogrammed human duct cells (Lee et al. eLife213; 2:e00940). These observations are discussed in our revised manuscript.

On the other hand, we cannot rule out the possibility that INS+ that we considered negative for the examined TF, actually contained small amounts of exogenous and/or endogenous protein that escaped our analysis. It should be emphasized that the assessment of staining extent using typical IHC techniques can be fairly subjective and is confounded by the existence of high and low signals in the same preparation, the latter of which are challenging to accurately distinguish from background. Reported co-expression percentages of TF and INS in human islets are quite variable. As an example, using immunostaining, it was estimated that 60% and 70% of INS+ cells were positive for MAFA and PDX1 in human islets respectively (Guo et al. J Clin Invest 123: 3306-3316, 2013). For MAFA, this proportion is significantly smaller in human juvenile islets (Cyphert et al. Diabetes 68:337-348, 2019). Conversely, FACS sorting revealed lower co-expression of INS with Pdx1 (40%) and Nkx2-2 (60%) in beta cells originating from immature stem cells as opposed to more differentiated cells (90%) (Nair et al. Nat Cell Biol 21, 263-274, 2019). Together, these observations support that 5TF cells are not fully mature. Ongoing work in our lab is trying to refine the current TF-based reprogramming protocol to achieve a higher degree of differentiation of the converted cells.

6. To have a better understanding of molecular similarities and differences between fibroblast-derived beta-like cells with pancreatic beta cells, gene profiling is much preferred over qPCR of a small number of genes.

Response: Following the reviewer advice we have performed gene profiling. RNA-seq data is now shown in Fig.5 of the revised manuscript.

7. Foreskin and adult fibroblasts were mentioned in the Method but it is not clearly what data were generated from what cells.

Response: All the experiments in the submitted manuscript were performed with control human foreskin fibroblast. We acknowledge that we also mentioned fibroblasts from T1D patients inadvertently in the Methodology Section of the submitted transcript. This has been corrected in the revised manuscript.

REVIEWERS' COMMENTS:

Reviewer #1 (Remarks to the Author):

The manuscript is now substantially revised and shown more convincing characterization data. The inclusion of RNAseq studies and beta cell functional genes quantification has improved the quality and importance of this study. Despite few technical limitations, I am convinced that it is now suitable for publication in its current form.

Reviewer #3 (Remarks to the Author):

The authors have improved the manuscript by clarification and addition of new data. I support publication of this study. A few minor points to consider below.

Page 17, top paragraph. I suggest to rephrase "...notoriously reduced...". Also, it is not clear what the authors mean by "indicating that cell loss was not exclusive of reprogramming INS+ cells".
Fig.S4. Somatostatin staining is difficult to see in yellow, suggest changing to a different color.
Fig.S9B. I assume day 0 and day 20 data are paired? If so, it would be informative to know the changes over time by linking the paired data points with lines.

REBUTTAL LETTER

COMMSBIO-21-3291B

Direct reprogramming of human fibroblasts into insulin-producing cells using transcription factors

We thank the reviewers for their thorough scientific review of our work. Please see responses below.

Reviewer #1 (Remarks to the Author):

The manuscript is now substantially revised and shown more convincing characterization data. The inclusion of RNAseq studies and beta cell functional genes quantification has improved the quality and importance of this study. Despite few technical limitations, I am convinced that it is now suitable for publication in its current form.

Response: we are pleased that the reviewer finds our manuscript suitable for publication.

Reviewer #3 (Remarks to the Author):

The authors have improved the manuscript by clarification and addition of new data. I support publication of this study. A few minor points to consider below.

1) Page 17, top paragraph. I suggest to rephrase "...notoriously reduced...". Also, it is not clear what the authors mean by "indicating that cell loss was not exclusive of reprogramming INS+ cells".

Response: We have re-phrased these sentences in the revised manuscript. Changes are copied below:

"...notoriously reduced..." changed to *"..reduced..."*

"indicating that cell loss was not exclusive of reprogramming INS+ cells" changed to *"demonstrating the maintenance of reprogramming"*

2) Fig.S4. Somatostatin staining is difficult to see in yellow, suggest changing to a different color.

Response: We have changed the color scheme of all figures of the manuscript to avoid the use of red/green in the same image. In new Figure S3 (old Figure S4), insulin is now shown in yellow, glucagon in red and somatostatin in turquoise blue.

3) Fig.S9B. I assume day 0 and day 20 data are paired? If so, it would be informative to know the changes over time by linking the paired data points with lines.

Response: Yes, data are paired. We now show individual responses in new Figure S8 panel c (old figure S9). Each mouse is depicted with a different color or style line.